# BDS-3/Galileo Time and Frequency Transfer with Quad-Frequency Precise Point Positioning

Yulong Ge [1], Xinyun Cao [2,3,4,*], Fei Shen [2,3,4] , Xuhai Yang [5,6] and Shengli Wang [7]

1 School of Marine Science and Engineering, Nanjing Normal University, Nanjing 210023, China; geyulong15@mials.ucas.ac.cn
2 School of Geography, Nanjing Normal University, Nanjing 210023, China; shen.f@njnu.edu.cn
3 Key Laboratory of Virtual Geographic Environment (Nanjing Normal University), Ministry of Education, Nanjing 210023, China
4 Jiangsu Center for Collaborative Innovation in Geographical Information Resource Development and Application, Nanjing 210023, China
5 National Time Service Center, Chinese Academy of Sciences, Xi'an 710600, China; yyang@ntsc.ac.cn
6 School of Astronomy and Space Science, University of Chinese Academy of Sciences, Beijing 100049, China
7 Collage of Ocean Science and Engineering, Shandong University of Science and Technology, Qingdao 266000, China; shlwang@sdust.edu.cn
* Correspondence: xycao@njnu.edu.cn; Tel.: +86-02-58-598-565

**Abstract:** In this work, quad-frequency precise point positioning (PPP) time and frequency transfer methods using Galileo E1/E5a/E5b/E5 and BDS-3 B1I/B3I/B1C/B2a observations were proposed with corresponding mathematical models. In addition, the traditional dual-frequency (BDS-3 B1I/B3I and Galileo E1/E5a) ionospheric-free (IF) model was also described and tested for comparison. To assess the proposed method for time transfer, datasets selected from timing labs were utilized and tested. Moreover, the number of Galileo or BDS-3 satellites, pseudorange residuals, positioning accuracy and tropospheric delay at receiver end were all analyzed. The results showed that the proposed quad-frequency BDS-3 or Galileo PPP models could be used to time transfer, due to stability and accuracy identical to that of dual-frequency IF model. Furthermore, the quad-frequency models can provide potential for enhancing the reliability and redundancy compared to the dual-frequency time transfer method.

**Keywords:** BDS-3; Galileo; quad-frequency; time and frequency transfer; PPP

## 1. Introduction

Following the widely application of GPS and GLONASS system, for example, precise point positioning (PPP) [1,2], ionosphere estimations [3], zenith tropospheric delay (ZTD) estimations [4] and time/frequency transfer [5,6], Galileo and BDS system have drawn a lot of interest. Galileo satellite system has provided initial services since December 15, 2016 [7]. By November 11, 2019, 24 Galileo satellites was launched successfully. BDS system was divided into three steps. The second-generation constellation, namely BDS-2, which contains 14 satellites, has provided the regional service since 2012 [8]. For third generation, namely BDS-3, was announced to provide a global service by December 2018. Now, the entire BDS-3 constellation, which consists of 3 GEO, 3 IGSO and 24 MEO satellites (https://www.glonass-iac.ru/en/BEIDOU/, accessed on 8 July 2021) [9].

Currently, numerous studies focus on Galileo [10] and BDS-3 positioning performance [11–13]. The positioning accuracy using pseudorange observations is better 10 m for Galileo-only [10] and BDS-3-only [11,12]. For high-precise positioning, for example, PPP and real-time kinematic (RTK) positioning, the accuracy all can reach the centimeter level [10,14–16]. By the end of 2018, the timing accuracy of BDS-3 has officially announced to reach better than 20 ns [11]. For Galileo timing service, the maximum tolerable error (MTE) of service levels 3 is about 10 ns for the current and needs of users [17].

Unlike GPS and GLONASS, all BDS-3 and Galileo satellites can transmit five signals' services. Users can receive open service signals about BDS-3 B1I/B3I/B1C/B2a and about Galileo E1/E5a/E5b/E5. At present, GNSS applications with dual- and triple-frequency observations were studied by many researches [18,19]. GNSS application with triple-frequency observation presented an obviously benefit for ambiguity resolution (AR) [20], cycle detection and time transfer [21] in contrast with dual-frequency observation, the reliability of positioning [22]. Besides, Galileo quad-frequency RTK models was studied by Tu, et al. [23]. Multi-frequency observations present many inherent advantages in GNSS application. Thus, we expect that PPP time transfer with Galileo or BDS-3 quad-frequency observations will show more superiority.

Nowadays, GPS PPP technique has been usually applied for international time comparison in time community [6,24]. By the development of multi-GNSS, PPP with multi-GNSS observation has achieved a hot topic for time comparison. Ge, et al. [25] presented GLONASS PPP time transfer with inter-frequency code biases (IFCBs) model. BDS PPP using triple-frequency observation was investigated by Tu, Zhang, Zhang, Liu and Lu [21]. In addition, multi-GNSS PPP time transfer adding the receiver clock model was studied by Ge, et al. [26]. Nevertheless, the above studies focused on dual- or triple-frequency PPP time transfer and are of no investigation to quad-frequency PPP time transfer. The use of Galileo and BDS-3 quad-frequency observations for PPP time transfer present greater challenges. Therefore, Galileo and BDS-3 quad-frequency observations are anticipated to be useful in improving the performance of PPP time transfer.

With this background, quad-frequency PPP models with BDS-3 B1I/B3I/B1C/B2a signals and Galileo E1/E5a/E5b/E5 signals are proposed and assessed. In this work, the quad-frequency PPP time transfer models of BDS-3 and Galileo are first presented. Experimental data used are then introduced. Finally, the conclusions are given.

## 2. Methods

In this section, we shall start with general observations models for BDS-3 or Galileo quad-frequency signals. Following is that four BDS-3 or Galileo PPP time transfer models are then proposed. We end with detailed analysis of four different models.

### 2.1. General Observations

The quad-frequency un-combined pseudorange and carrier phase observations can be written as [27,28]

$$\begin{cases} p_{r,1}^s = \boldsymbol{g}_r^s \cdot \boldsymbol{x} + cdt_r + m_r^s \cdot Z_w + I_{r,1}^s + \beta_{12} \cdot c\mathrm{DCB}_{12}^s + cd_{r,1} + \varepsilon_{r,1}^s \\ p_{r,2}^s = \boldsymbol{g}_r^s \cdot \boldsymbol{x} + cdt_r + m_r^s \cdot Z_w + \gamma_2 \cdot I_{r,1}^s - \alpha_{12} \cdot c\mathrm{DCB}_{12}^s + cd_{r,2} + \varepsilon_{r,2}^s \\ p_{r,3}^s = \boldsymbol{g}_r^s \cdot \boldsymbol{x} + cdt_r + m_r^s \cdot Z_w + \gamma_3 \cdot I_{r,1}^s - cd_{\mathrm{IF}_{12}}^s + c(d_{r,3} + d_3^s) + \varepsilon_{r,3}^s \\ p_{r,4}^s = \boldsymbol{g}_r^s \cdot \boldsymbol{x} + cdt_r + m_r^s \cdot Z_w + \gamma_4 \cdot I_{r,1}^s - cd_{\mathrm{IF}_{12}}^s + c(d_{r,4} + d_4^s) + \varepsilon_{r,4}^s \end{cases} \quad (1)$$

$$\begin{cases} l_{r,1}^s = \boldsymbol{g}_r^s \cdot \boldsymbol{x} + cdt_r + m_r^s \cdot Z_w - I_{r,1}^s - cd_{\mathrm{IF}_{12}}^s + \lambda_1(N_{r,1}^s + b_{r,1} + b_1^s) + \xi_{r,1}^s \\ l_{r,2}^s = \boldsymbol{g}_r^s \cdot \boldsymbol{x} + cdt_r + m_r^s \cdot Z_w - \gamma_2 \cdot I_{r,1}^s - cd_{\mathrm{IF}_{12}}^s + \lambda_2(N_{r,2}^s + b_{r,2} + b_2^s) + \xi_{r,2}^s \\ l_{r,3}^s = \boldsymbol{g}_r^s \cdot \boldsymbol{x} + cdt_r + m_r^s \cdot Z_w - \gamma_3 \cdot I_{r,1}^s - cd_{\mathrm{IF}_{12}}^s + \lambda_3(N_{r,3}^s + b_{r,3} + b_3^s) + \xi_{r,3}^s \\ l_{r,4}^s = \boldsymbol{g}_r^s \cdot \boldsymbol{x} + cdt_r + m_r^s \cdot Z_w - \gamma_4 \cdot I_{r,1}^s - cd_{\mathrm{IF}_{12}}^s + \lambda_4(N_{r,4}^s + b_{r,4} + b_4^s) + \xi_{r,4}^s \end{cases} \quad (2)$$

$$\begin{cases} \alpha_{mn} = \frac{(f_m^s)^2}{(f_m^s)^2 - (f_n^s)^2}, \ \beta_{mn} = -\frac{(f_n^s)^2}{(f_m^s)^2 - (f_n^s)^2} \\ \mathrm{DCB}_{mn}^s = d_m^s - d_n^s, \ \mathrm{DCB}_{r,mn}^s = d_{r,m}^s - d_{r,n}^s \\ d_{\mathrm{IF}_{mn}}^s = \alpha_{mn} \cdot d_m^s + \beta_{mn} \cdot d_n^s, \ d_{r,\mathrm{IF}_{mn}}^s = \alpha_{mn} \cdot d_{r,m}^s + \beta_{mn} \cdot d_{r,n}^s \end{cases} \quad (3)$$

where $S$ and $r$ represent satellite and receiver, respectively; 1, 2, 3, and 4 are BDS-3 (B1I/B3I/B1C/B2a) or Galileo (E1/E5a/E5b/E5) quad-frequency signals. $l$ and $p$ are the observed-minus-computed (OMC) values of carrier-phase and pseudorange observations, respectively. $\boldsymbol{g}_r^s$ is the unit vector; $\boldsymbol{x}$ refers to the vector of the receiver coordinate increments in three components. $c$ indicates the speed of light; $dt_r$ is the receiver clock

offset; $m_r^s$ indicates the wet mapping function; $Z_w$ demonstrates the zenith wet delay; $\gamma_j$ is the frequency-dependent multiplier factor ($\gamma_j = (f_1/f_j)^2, j = 2, 3, 4$). $I_{r,1}^s$ is the slant ionospheric delay on the first frequency $f_1^s$. $\alpha_{mn}$ and $\beta_{mn}$ are the frequency factors ($m \neq n$). $d_m^s$ and $d_{r,m}$ are the uncalibrated code bias (UCD) ($m = 1, 2, 3, 4$) at satellite and receiver end, respectively; $d_{\mathrm{IF}_{mn}}^s$ is the ionospheric-free UCD at satellite $s$. $\mathrm{DCB}_{mn}^s$ and $\mathrm{DCB}_{r,mn}^s$ are the differential code bias at satellite and receiver end. $\lambda_j$ refers to the wavelength; $N_{r,j}^s$ indicates the integer ambiguity; $b_{r,j}$ and $b_j^s$ are phase delay at the receiver and satellite end, respectively. $\varepsilon_{r,j}^s$ and $\zeta_{r,j}^s$ are the measurement noise of pseudorange and carrier phase observations. Here, $m$ and $n$ are the different frequency. In addition, in order to keep the receiver offset in multi-frequency PPP model consistent with that of dual-frequency, we use the receiver offset in dual-frequency ionospheric-free (IF) model as a reference.

*2.2. Dual-Frequency IF PPP Model*

The dual-frequency IF PPP is commonly employed for PPP model. Here, BDS-3 (B1I and B3I) or Galileo (E1 and E5a) were utilized to generate the dual-frequency IF PPP model, which is called IF0 in our study. The IF0 model can be expressed as [27,29]:

$$
\begin{cases}
p_{r,\mathrm{IF12}}^s = \boldsymbol{g}_r^s \cdot \boldsymbol{x} + cdt_{r,\mathrm{IF12}} + m_r^s \cdot Z_w + \varepsilon_{r,\mathrm{IF12}}^s \\
l_{r,\mathrm{IF12}}^s = \boldsymbol{g}_r^s \cdot \boldsymbol{x} + cdt_{r,\mathrm{IF12}} + m_r^s \cdot Z_w + \overline{N}_{r,\mathrm{IF12}}^s + \zeta_{r,\mathrm{IF12}}^s
\end{cases}
\tag{4}
$$

with

$$
\begin{cases}
cdt_{r,\mathrm{IF12}} = cdt_r + cd_{r,\mathrm{IF12}} \\
\overline{N}_{r,\mathrm{IF12}}^s = \alpha_{12}\lambda_1(N_{r,1}^s + b_{r,1} + b_1^s) + \beta_{12}\lambda_2(N_{r,2}^s + b_{r,2} + b_2^s) - c(d_{r,\mathrm{IF12}} + d_{\mathrm{IF12}}^s) \\
\varepsilon_{r,\mathrm{IF12}}^s = \alpha_{12}\varepsilon_{r,1}^s + \beta_{12}\varepsilon_{r,2}^s \\
\zeta_{r,\mathrm{IF12}}^s = \alpha_{12}\zeta_{r,1}^s + \beta_{12}\zeta_{r,2}^s
\end{cases}
\tag{5}
$$

where $p_{r,\mathrm{IF12}}^s$ and $l_{r,\mathrm{IF12}}^s$ are the IF combination pseudorange and carrier phase OMC values, respectively; $cdt_{r,\mathrm{IF12}}^s$ is the receiver clock offset, which has absorbed the ionospheric-free combination UCD at receiver end; $N_{r,\mathrm{IF12}}^s$ indicates the float ambiguity; $\varepsilon_{r,\mathrm{IF12}}^s$ and $\zeta_{r,\mathrm{IF12}}^s$ are the ionospheric-free measurement noise. Here, 1 and 2 represent BDS-3 (B1I, B1I) and Galileo (E1, E5a).

The estimated parameters of IF0 model can be expressed as

$$
\boldsymbol{S} = [\boldsymbol{x}, cdt_{r,\mathrm{IF12}}, Z_w, \overline{N}_{r,\mathrm{IF12}}^s]
\tag{6}
$$

*2.3. Quad-Frequency IF PPP Models*

Usually, three dual-frequency IF PPP model can be obtained by Galileo E1, E5a, E5b, and E5. However, the noise amplification of E5a, E5b, and E5 combination is obviously larger than E1/E5a, E1/E5b, and E1/E5. Hence, for quad-frequency Galileo IF PPP, which is called IF1 (Galileo) in our work, with three dual-frequency IF model combinations (E1/E5a, E1/E5b and E1/E5) is considered here.

Unlike IF0, three dual-frequency IF model will generate three receiver clock offsets. Note that the receiver clock offset has absorbed the UCD at receiver end (see Equation (3)). In order to separate $cdt_{r,\mathrm{IF12}}$, the receiver clock of E1/E5b and E1/E5 combination will be divided into $cdt_{r,\mathrm{IF12}}$ and an inter-frequency (IFB) parameter. The method will simplify measurement of UCDs for different frequency at receiver end. User just calibrates the UCD of IF0 model at receiver end. Then, the linearized observation equations can be expressed as [30]:

$$
\begin{cases}
p_{r,\mathrm{IF12}}^s = \boldsymbol{g}_r^s \cdot \boldsymbol{x} + cdt_{r,\mathrm{IF12}} + m_r^s \cdot Z_w + \varepsilon_{r,\mathrm{IF12}}^s \\
p_{r,\mathrm{IF13}}^s = \boldsymbol{g}_r^s \cdot \boldsymbol{x} + cdt_{r,\mathrm{IF12}} + m_r^s \cdot Z_w + \Omega_{r,\mathrm{IF13}}^s + \varepsilon_{r,\mathrm{IF13}}^s \\
p_{r,\mathrm{IF14}}^s = \boldsymbol{g}_r^s \cdot \boldsymbol{x} + cdt_{r,\mathrm{IF12}} + m_r^s \cdot Z_w + \Omega_{r,\mathrm{IF14}}^s + \varepsilon_{r,\mathrm{IF14}}^s
\end{cases}
\tag{7}
$$

$$\begin{cases} l^s_{r,\text{IF12}} = \boldsymbol{g}^s_r \cdot \boldsymbol{x} + cdt_{r,\text{IF12}} + m^s_r \cdot Z_w + \overline{N}^s_{r,\text{IF12}} + \xi^s_{r,\text{IF12}} \\ l^s_{r,\text{IF13}} = \boldsymbol{g}^s_r \cdot \boldsymbol{x} + cdt_{r,\text{IF12}} + m^s_r \cdot Z_w + \overline{N}^s_{r,\text{IF13}} + \xi^s_{r,\text{IF13}} \\ l^s_{r,\text{IF14}} = \boldsymbol{g}^s_r \cdot \boldsymbol{x} + cdt_{r,\text{IF12}} + m^s_r \cdot Z_w + \overline{N}^s_{r,\text{IF14}} + \xi^s_{r,\text{IF14}} \end{cases} \tag{8}$$

$$\begin{cases} \Omega^s_{r,\text{IF13}} = c(\beta_{12}\text{DCB}_{r,12} - \beta_{13}\text{DCB}_{r,13}) + c(\beta_{12}\text{DCB}^s_{12} - \beta_{13}\text{DCB}^s_{13}) \\ \Omega^s_{r,\text{IF14}} = c(\beta_{12}\text{DCB}_{r,12} - \beta_{14}\text{DCB}_{r,14}) + c(\beta_{12}\text{DCB}^s_{12} - \beta_{14}\text{DCB}^s_{14}) \\ \overline{N}^s_{r,\text{IF12}} = \alpha_{12}\lambda_1(N^s_{r,1} + b_{r,1} + b^s_1) + \beta_{12}\lambda_2(N^s_{r,2} + b_{r,2} + b^s_2) - c(d_{r,\text{IF12}} + d^s_{\text{IF12}}) \\ \overline{N}^s_{r,\text{IF13}} = \alpha_{13}\lambda_1(N^s_{r,1} + b_{r,1} + b^s_1) + \beta_{13}\lambda_3(N^s_{r,3} + b_{r,3} + b^s_3) - c(d_{r,\text{IF12}} + d^s_{\text{IF12}}) \\ \overline{N}^s_{r,\text{IF14}} = \alpha_{14}\lambda_1(N^s_{r,1} + b_{r,1} + b^s_1) + \beta_{14}\lambda_4(N^s_{r,4} + b_{r,4} + b^s_4) - c(d_{r,\text{IF12}} + d^s_{\text{IF12}}) \end{cases} \tag{9}$$

where $p^s_{r,\text{IF13}}, p^s_{r,\text{IF14}}, l^s_{r,\text{IF13}}$, and $l^s_{r,\text{IF14}}$ are the IF combination pseudorange and carrier phase OMC values, respectively; $\Omega^s_{r,\text{IF13}}$ and $\Omega^s_{r,\text{IF14}}$ are the IFB for E1/E5b and E1/E5 IF combinations, respectively; $\overline{N}^s_{r,\text{IF13}}$ and $\overline{N}^s_{r,\text{IF14}}$ are the redefinition ambiguity for E1/E5b and E1/E5 IF combinations, respectively. Note that 1, 2, 3, and 4 represent E1, E5a, E5b and E5 signals for Galileo.

Then, the estimated parameters of IF1 (Galileo) can be written as

$$\boldsymbol{S} = [\boldsymbol{x}, cdt_{r,\text{IF12}}, Z_w, \Omega^s_{r,\text{IF13}}, \Omega^s_{r,\text{IF14}}, \overline{N}^s_{r,\text{IF12}}, \overline{N}^s_{r,\text{IF13}}, \overline{N}^s_{r,\text{IF14}}] \tag{10}$$

For BDS-3, two dual-frequency combination (B1I/B3I and B1C/B2a) are recommended by the Interface Control Document (ICD) [31]. As we point out, an IFB parameter needs to be added. Thus, quad-frequency BDS-3 IF model, called IF1 (BDS-3) in our study, will generate by two dual-frequency IF combinations, and can be expressed as

$$\begin{cases} p^s_{r,\text{IF12}} = \boldsymbol{g}^s_r \cdot \boldsymbol{x} + cdt_{r,\text{IF12}} + m^s_r \cdot Z_w + \varepsilon^s_{r,\text{IF12}} \\ p^s_{r,\text{IF34}} = \boldsymbol{g}^s_r \cdot \boldsymbol{x} + cdt_{r,\text{IF12}} + m^s_r \cdot Z_w + \Omega^s_{r,\text{IF34}} + \varepsilon^s_{r,\text{IF34}} \end{cases} \tag{11}$$

$$\begin{cases} l^s_{r,\text{IF12}} = \boldsymbol{g}^s_r \cdot \boldsymbol{x} + cdt_{r,\text{IF12}} + m^s_r \cdot Z_w + \overline{N}^s_{r,\text{IF12}} + \xi^s_{r,\text{IF12}} \\ l^s_{r,\text{IF34}} = \boldsymbol{g}^s_r \cdot \boldsymbol{x} + cdt_{r,\text{IF12}} + m^s_r \cdot Z_w + \overline{N}^s_{r,\text{IF34}} + \xi^s_{r,\text{IF34}} \end{cases} \tag{12}$$

$$\begin{cases} \Omega^s_{r,\text{IF34}} = c(d_{r,\text{IF34}} - d_{r,\text{IF12}}) + c(\frac{\alpha_{34}\beta_{12}}{\beta_{13}} + \frac{\beta_{34}\beta_{12}}{\beta_{14}})\text{DCB}^s_{12} - \alpha_{34}\text{DCB}^s_{13} - \beta_{34}\text{DCB}^s_{14} \\ \overline{N}^s_{r,\text{IF34}} = \alpha_{34}\lambda_3(N^s_{r,3} + b_{r,3} + b^s_3) + \beta_{34}\lambda_4(N^s_{r,4} + b_{r,4} + b^s_4) - c(d_{r,\text{IF12}} + d^s_{\text{IF12}}) \end{cases} \tag{13}$$

where $p^s_{r,\text{IF34}}$ and $l^s_{r,\text{IF34}}$ are the B1C/B2a IF combination pseudorange and carrier phase OMC values, respectively; $\Omega^s_{r,\text{IF34}}$ is the IFB for B1C/B2a IF combination; $\overline{N}^s_{r,\text{IF34}}$ is the redefinition ambiguity for B1C/B2a IF combination; note further that 1, 2, 3, and 4 represent B1I, B3I, B1C, and B2a signals for BDS-3 here.

Then, the estimated parameters of IF1 (BDS-3) can be written as

$$\boldsymbol{S} = [\boldsymbol{x}, cdt_{r,\text{IF12}}, Z_w, \Omega^s_{r,\text{IF34}}, \overline{N}^s_{r,\text{IF34}}] \tag{14}$$

Quad-frequency IF model, namely IF2, can also be generated by two triple-frequency combinations. The three combination coefficients need to meet three conditions (see Equation (5)), consists of unchanged geometric range, elimination of first-order ionospheric delays, and minimum noise [32].

$$\begin{cases} e_1 + e_2 + e_3 = 1 \\ e_1\gamma_1 + e_2\gamma_2 + e_3\gamma_3 = 0 \\ e_1^2 + e_2^2 + e_3^2 = \varepsilon^2 = \min \end{cases} \tag{15}$$

where $e_1$, $e_2$, and $e_3$ are the combination coefficients. $e_1$, $e_2$, and $e_3$ can be calculated by the Lagrange Equation [18], the results can be described as [33]:

$$\begin{cases} e_1 = \dfrac{\gamma_2^2 + \gamma_3^2 - \gamma_2 - \gamma_3}{2 \cdot (\gamma_2^2 + \gamma_3^2 - \gamma_2 \cdot \gamma_3 - \gamma_3 - \gamma_2 + 1)} \\[2mm] e_2 = \dfrac{\gamma_3^2 - \gamma_2 \cdot \gamma_3 - \gamma_2 + 1}{2 \cdot (\gamma_2^2 + \gamma_3^2 - \gamma_2 \cdot \gamma_3 - \gamma_3 - \gamma_2 + 1)} \\[2mm] e_3 = \dfrac{\gamma_2^2 - \gamma_2 \cdot \gamma_3 - \gamma_3 + 1}{2 \cdot (\gamma_2^2 + \gamma_3^2 - \gamma_2 \cdot \gamma_3 - \gamma_3 - \gamma_2 + 1)} \end{cases} \tag{16}$$

An IFB parameter should be added in IF2 model. Then, the IF2 model for BDS-3 (B1I/B3I/B1C and B1I/B3I/B1C) or Galileo (E1/E5a/E5b and E1/E5a/E5) can be written as [32]:

$$\begin{cases} p_{r,\text{IF123}}^s = e_1 p_{r,1}^s + e_2 p_{r,2}^s + e_3 p_{r,3}^s = g_r^s \cdot x + cdt_{r,\text{IF123}} + m_r^s \cdot Z_w + \varepsilon_{r,\text{IF123}}^s \\ p_{r,\text{IF124}}^s = e_{11} p_{r,1}^s + e_{22} p_{r,2}^s + e_{44} p_{r,4}^s = g_r^s \cdot x + cdt_{r,\text{IF123}} + m_r^s \cdot Z_w + \Omega_{\text{IF124}}^s + \varepsilon_{r,\text{IF124}}^s \end{cases} \tag{17}$$

$$\begin{cases} l_{r,\text{IF123}}^s = e_1 l_{r,1}^s + e_2 l_{r,2}^s + e_3 l_{r,3}^s = g_r^s \cdot x + cdt_{r,\text{IF123}} + m_r^s \cdot Z_w + N_{r,\text{IF123}}^s + \xi_{r,\text{IF123}}^s \\ l_{r,\text{IF124}}^s = e_{11} l_{r,1}^s + e_{22} l_{r,2}^s + e_{44} l_{r,4}^s = g_r^s \cdot x + cdt_{r,\text{IF123}} + m_r^s \cdot Z_w + N_{r,\text{IF124}}^s + \xi_{r,\text{IF124}}^s \end{cases} \tag{18}$$

$$\begin{cases} cdt_{r,\text{IF123}} &= cdt_{r,\text{IF12}} + c[(\beta_{12} - e_2)\text{DCB}_{r,12} - e_3\text{DCB}_{r,13}] \\ \Omega_{\text{IF124}}^s &= e_3\text{DCB}_{r,13} - e_{44}\text{DCB}_{r,14} \\ N_{r,\text{IF123}}^s &= e_1\lambda_1(N_{r,1}^s + b_{r,1} + b_1^s) + e_2\lambda_2(N_{r,2}^s + b_{r,2} + b_2^s) \\ &\quad + e_3\lambda_3(N_{r,3}^s + b_{r,3} + b_3^s) - c(d_{r,\text{IF12}} + d_{\text{IF12}}^s) \\ &\quad - c[(\beta_{12} - e_2)\text{DCB}_{r,12} - e_3\text{DCB}_{r,13}] \\ N_{r,\text{IF124}}^s &= e_{11}\lambda_1(N_{r,1}^s + b_{r,1} + b_1^s) + e_{22}\lambda_2(N_{r,2}^s + b_{r,2} + b_2^s) \\ &\quad + e_{44}\lambda_4(N_{r,4}^s + b_{r,4} + b_4^s) - c(d_{r,\text{IF12}} + d_{\text{IF12}}^s) \\ &\quad - c[(\beta_{12} - e_2)\text{DCB}_{r,12} - e_{44}\text{DCB}_{r,14}] \end{cases} \tag{19}$$

where $cdt_{r,\text{IF123}}$ refers to the receiver clock; $\Omega_{r,\text{IF124}}^s$ is the IFB parameter; $N_{r,\text{IF123}}^s$ and $N_{r,\text{IF124}}^s$ are the redefinition ambiguity for two triple-frequency IF models, respectively. $e_{11}$, $e_{22}$, and $e_{44}$ is the combination coefficients.

The estimates parameters of IF2 can be obtained as

$$S = [x, cdt_{r,\text{IF123}}, Z_w, \Omega_{r,\text{IF124}}^s, \overline{N}_{r,\text{IF123}}^s, \overline{N}_{r,\text{IF124}}^s] \tag{20}$$

Not that 1, 2, 3, and 4 represent B1I, B3I, B1C, and B2a signals for BDS-3 or E1, E5a, E5b, and E5 for Galileo here.

### 2.4. Quad-Frequency Uncombined PPP Model

For quad-frequency uncombined PPP model, namely UC in this work, the slant ionospheric delay will be generally estimated as a parameter. Combined Equations (1) and (2), the precise quad-frequency uncombined PPP model can be described as [29,30]:

$$\begin{cases} p_{r,1}^s = g_r^s \cdot x + cdt_{r,\text{IF12}} + m_r^s \cdot Z_w + I_{r,\text{UC1234}}^s + \varepsilon_{r,1}^s \\ p_{r,2}^s = g_r^s \cdot x + cdt_{r,\text{IF12}} + m_r^s \cdot Z_w + \gamma_2 \cdot I_{r,\text{UC1234}}^s + \varepsilon_{r,2}^s \\ p_{r,3}^s = g_r^s \cdot x + cdt_{r,\text{IF12}} + m_r^s \cdot Z_w + \gamma_3 \cdot I_{r,\text{UC1234}}^s + \Omega_{r,3}^s + \varepsilon_{r,3}^s \\ p_{r,4}^s = g_r^s \cdot x + cdt_{r,\text{IF12}} + m_r^s \cdot Z_w + \gamma_4 \cdot I_{r,\text{UC1234}}^s + \Omega_{r,4}^s + \varepsilon_{r,4}^s \end{cases} \tag{21}$$

$$\begin{cases} l_{r,1}^s = g_r^s \cdot x + cdt_{r,\text{IF12}} + m_r^s \cdot Z_w - I_{r,\text{UC1234}}^s + \overline{N}_{r,1}^s + \xi_{r,1}^s \\ l_{r,2}^s = g_r^s \cdot x + cdt_{r,\text{IF12}} + m_r^s \cdot Z_w - \gamma_2 \cdot I_{r,\text{UC1234}}^s + \overline{N}_{r,2}^s + \xi_{r,2}^s \\ l_{r,3}^s = g_r^s \cdot x + cdt_{r,\text{IF12}} + m_r^s \cdot Z_w - \gamma_3 \cdot I_{r,\text{UC1234}}^s + \overline{N}_{r,3}^s + \xi_{r,3}^s \\ l_{r,4}^s = g_r^s \cdot x + cdt_{r,\text{IF12}} + m_r^s \cdot Z_w - \gamma_4 \cdot I_{r,\text{UC1234}}^s + \overline{N}_{r,4}^s + \xi_{r,4}^s \end{cases} \tag{22}$$

$$
\begin{cases}
cdt_{r,\text{IF12}} = cdt_r + cd_{r,\text{IF12}} \\
\Omega^s_{r,3} = c(\frac{\beta_{12}}{\beta_{13}}\text{DCB}_{r,12} - \text{DCB}_{r,13}) + c(\frac{\beta_{12}}{\beta_{13}}\text{DCB}^s_{12} - \text{DCB}^s_{13}) \\
\Omega^s_{r,4} = c(\frac{\beta_{12}}{\beta_{14}}\text{DCB}_{r,12} - \text{DCB}_{r,14}) + c(\frac{\beta_{12}}{\beta_{14}}\text{DCB}^s_{12} - \text{DCB}^s_{14}) \\
I^s_{r,\text{UC1234}} = I^s_{r,1} + \beta_{12}c(\text{DCB}_{r,12} + \text{DCB}^s_{12}) \\
\overline{N}^s_{r,j} = \lambda_j(N^s_{r,j} + b_{r,j} + b^s_j) - c(d_{r,\text{IF12}} + d^s_{\text{IF12}}) + \gamma_j\beta_{12}c(\text{DCB}_{r,12} + \text{DCB}^s_{12}) \\
j = 1, 2, 3, 4
\end{cases}
\tag{23}
$$

where $\Omega^s_{r,3}$ and $\Omega^s_{r,4}$ are the corresponding IFB parameters; $I^s_{r,\text{UC1234}}$ and $\overline{N}^s_{r,j}$ are the redefined slant ionospheric delay and float ambiguity parameters, respectively. Then, the unknown parameters of UC model can be expressed as:

$$
S = [x, cdt_{r,\text{IF12}}, Z_w, \Omega^s_{r,3}, \Omega^s_{r,4}, I^s_{r,\text{UC1234}}, \overline{N}^s_{r,1}, \overline{N}^s_{r,2}, \overline{N}^s_{r,3}, \overline{N}^s_{r,4}]
\tag{24}
$$

*2.5. Characteristic of Quad-Frequency GNSS PPP Time Transfer Models*

To compare the IF0, IF1, IF2, and UC models with BDS-3 or Galileo observations, their major characteristics are concluded in Tables 1 and 2, respectively, including the observations used, the combination coefficients, and the noise amplification factor.

**Table 1.** Comparison of dual- and four- frequency BDS-3 PPP time transfer models.

|  |  | $e_1$ | $e_2$ | $e_3$ | $e_4$ | Noise Amplification |
|---|---|---|---|---|---|---|
| IF0 | B1I-B3I | 2.9437 | −1.9437 | 0 | 0 | 3.5275 |
| IF1 | B1I-B3I | 2.9437 | −1.9437 | 0 | 0 | 3.5275 |
|  | B1C-B2a | 0 | 0 | 2.2606 | −1.2606 | 2.5883 |
| IF2 | B1I-B3I-B1C | 1.3877 | −1.8908 | 1.5031 | 0 | 2.7857 |
|  | B1I-B3I-B2a | 2.3433 | −0.0893 | 0 | −1.2543 | 2.6594 |
| UC | B1I | 1 | 0 | 0 | 0 | 1 |
|  | B3I | 0 | 1 | 0 | 0 | 1 |
|  | B1C | 0 | 0 | 1 | 0 | 1 |
|  | B2a | 0 | 0 | 0 | 1 | 1 |

**Table 2.** Comparison of dual- and four- frequency Galileo PPP time transfer models.

|  |  | $e_1$ | $e_2$ | $e_3$ | $e_4$ | Noise Amplification |
|---|---|---|---|---|---|---|
| IF0 | E1-E5a | 2.2606 | −1.2606 | 0 | 0 | 2.5883 |
| IF1 | E1-E5a | 2.2606 | −1.2606 | 0 | 0 | 2.5883 |
|  | E1-E5b | 2.4220 | −1.4220 | 0 | 0 | 2.8086 |
|  | E1-E5 | 2.3380 | −1.3380 | 0 | 0 | 2.6938 |
| IF2 | E1-E5a-E5b | 2.3149 | −0.8363 | −0.4787 | 0 | 2.5075 |
|  | E1-E5a-E5 | 2.2929 | −0.7340 | 0 | −0.5589 | 2.4715 |
| UC | E1 | 1 | 0 | 0 | 0 | 1 |
|  | E5a | 0 | 1 | 0 | 0 | 1 |
|  | E5b | 0 | 0 | 1 | 0 | 1 |
|  | E5 | 0 | 0 | 0 | 1 | 1 |

From Table 1, we can see that the B1C/B2a presents smaller noise amplification than B1I/B3I or triple-frequency combination. In addition, the noise amplification of triple-frequency combination is better than that of B1I/B3I. Interestingly, B1I/B3I/B2a combination is analogous to B1I/B2a combination due to low contribution of B3I. For Galileo quad-frequency PPP time transfer models (see, Table 2), we can see that the noise amplification of E1/E5a, E1/E5b, and E1/E5 show the similar performance, while is slightly larger than E1/E5a/E5b and E1/E5a/E5. More interestingly, E1 presents the

greatest contribution for E1/E5a/E5b and E1/E5a/E5 combination. Note further that the combination coefficient retains at least 8 decimal points, otherwise it will result in an abnormal combined observation, although we keep only 4 decimal points in Table 2. The reader can calculate the combination coefficient by Equation (16).

On the other hand, the IF1 and UC models exhibit more flexible than the IF2 model. We do not recommend users to apply IF2 model for PPP time transfer. That can be explained by two reasons. For one thing, the IF2 model will be not used due to the absence of particular frequency. For another thing, users need to calibrate UCDs at three frequencies. As we know, the calibration of UCD is complex. For IF1 and UC model, UCD at two frequencies are only calibrated.

## 3. Results and Discussion

Processing strategies and datasets are first introduced. Subsequently, the quad-frequency BDS-3 and Galileo PPP time transfer solutions with different models are provided. Finally, the observation residuals, positioning, tropospheric delay, and the IFB are presented.

### 3.1. Data Acquisition and Processing Strategies

In our work, two stations (BRCH and XIA3) located in Physikalisch-Technische Bundesanstalt (PTB) in Braunschweig, Germany and National Time Service Center (NTSC) in Xi'an, China, from iGMAS (the international GNSS monitoring and assessment system), are selected for BDS-3 tests with a time span from DOY (day of year) 15–19, 2019. However, the three stations (BRUX, PT11 and PTBB), located in Royal Observatory of Belgium (ROB), Belgium and PTB, are chosen for Galileo tests with a time span from DOY 277–282, 2019. The information of the selected station is listed in Table 3. Note that high-performance H-master are available at all stations. The precise orbit and clock products (WUM) with intervals of 15 min and 30 s, respectively, are released by Wuhan University, China. The BDS-3- and Galileo-only quad-frequency PPP time transfer models are studied and analyzed. The detailed processing strategies for BDS-3- or Galileo-only quad-frequency PPP time transfer models are described in Table 4. Note that the GAMP [34] software was developed to meet the requirements of BDS-3 and Galileo quad-frequency PPP-derived time transfer experiments in our work.

**Table 3.** Information for selected station from timing lab.

|  | Station Name | Timing Lab | Receiver | Antenna | External Clock |
|---|---|---|---|---|---|
| BDS-3 | BRCH | PTB | CETC-54-GMR-4016 | GNSS-750 | UTC(PTB) |
|  | XIA3 | NTSC | CETC-54-GMR-4016 | NOV750.R4 | UTC(NTSC) |
| Galileo | PTBB | PTB | SEPT POLARX4TR | LEIAR25.R4—LEIT | UTC(PTB) |
|  | PT11 | PTB | SEPT POLARX5TR | LEIAR25.R4—LEIT | UTC(PTB) |
|  | BRUX | ROB | SEPT POLARX4TR | JAVRINGANT_DM—NONE | UTC(ROB) |

Figures 1 and 2 show the average distribution of time dilution of precision (TDOP) and number of satellites for GPS, GLONASS, Galileo, and BDS-3 on DOY 265–266, 2019. From two figures, we can conclude two interestingly findings. First, BDS-3 and Galileo can provide globe service from the TDOP values and the satellite number. Second, the average TDOP values and number of satellites are (0.62, 0.82, 0.94, 1.21) and (12.32, 8.47, 7.93, 6.32) for GPS, GLONASS, Galileo, and BDS-3, respectively. The number of BDS-3 satellites is a little less than that of Galileo at current state.

**Table 4.** Strategies for BDS-3 and Galileo IF0, IF1, IF2, and UC PPP time transfer model.

| Item | Strategy |
|---|---|
| Observations | BDS-3: undifferenced and uncombined observations in B1I, B3I, B1C, and B2a frequency<br>Galileo: undifferenced and uncombined observations in E1, E5a, E5b, and E5 frequency |
| Weight of observation | Elevation-dependent |
| Cutoff angle | 10° |
| Sampling rate | 30 s |
| PCO and PCV | Corrected |
| Station displacement | Corrected [35] |
| Wind up | Corrected [36] |
| Relativistic effect | Corrected |
| Estimator | Kalman filter |
| Phase ambiguity | Estimated as constants at each arc (float values) |
| IFB | Estimated as random walk (RW) noise [18] |
| Receiver position | Estimated as constants in static model |
| Receiver clock offset | Estimated as white noise |
| Ionospheric delay | IF0, IF1, IF2 models: first-order effects removed with IF combination<br>UC model: estimated as white noise |
| Tropospheric delay | Wet delay: estimated as random walk noise<br>Dry delay: corrected by Saastamoinen model<br>Mapping function: GMF [37] |

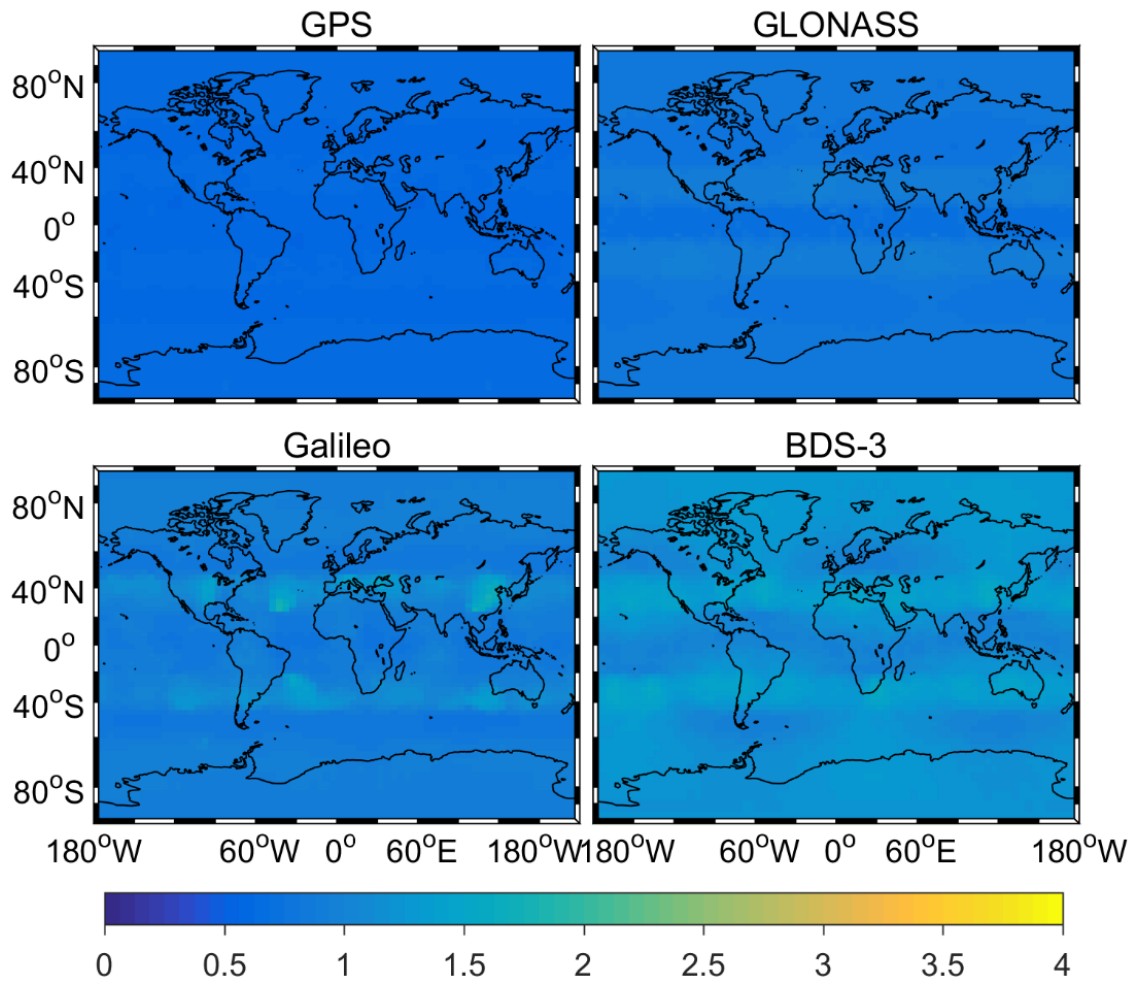

**Figure 1.** TDOP values of GPS, GLONASS, Galileo, and BDS-3.

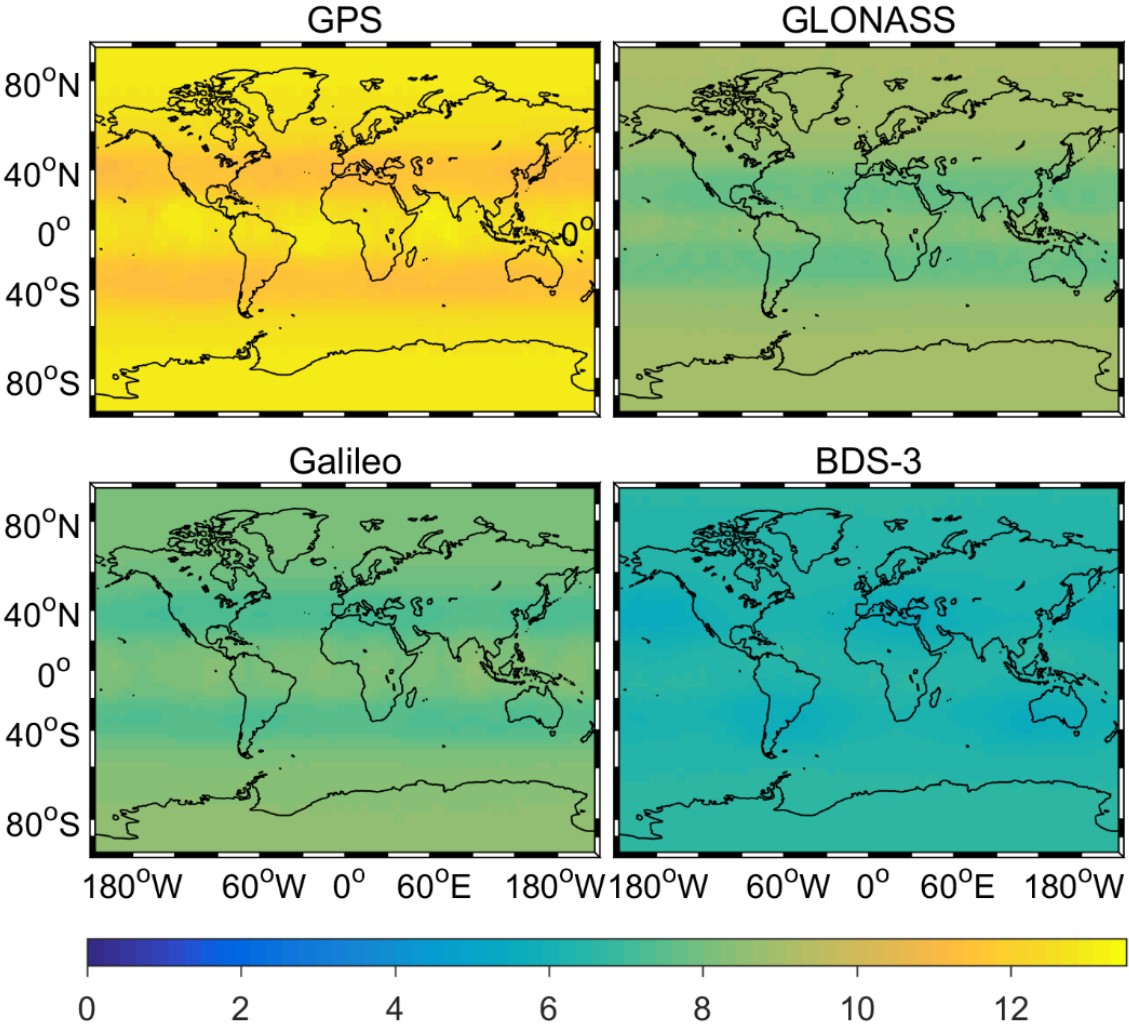

**Figure 2.** Number of satellites for GPS, GLONASS, Galileo, and BDS-3.

*3.2. Quad-Frequency BDS-3 PPP Time Transfer Solutions*

Figure 3 presents the time difference of XIA3-BRCH with BDS-3 IF0, IF1, IF2, and UC PPP models. Note that the difference has absorbed the hardware delay. The time series of IF0, IF1, and UC match each other very well. However, there is a clear system bias between IF2 and other models, which reflects the UCD at receiver end (see Equation (19)). In addition, the trends of time series obtained by different models are nearly consistent. For a clearer surface of our findings, we enlarge some of results in Figure 3 and present them in Figure 4. From Figure 4, we can further prove our previous findings. To further quality our results, from DOY 15 to 19, the standard deviation (STD) values are (0.54, 0.43, 0.47, 0.30, 0.60) ns, (0.49, 0.44, 0.46, 0.26, 0.62) ns, (0.59, 0.48, 0.51, 0.36, 0.64) ns, and (0.51, 0.44, 0.43, 0.26, 0.61) ns, respectively, for IF0, IF1, IF2, and UC models. Hence, we can conclude that four kinds of BDS-3 PPP time transfer models exhibit the same performance.

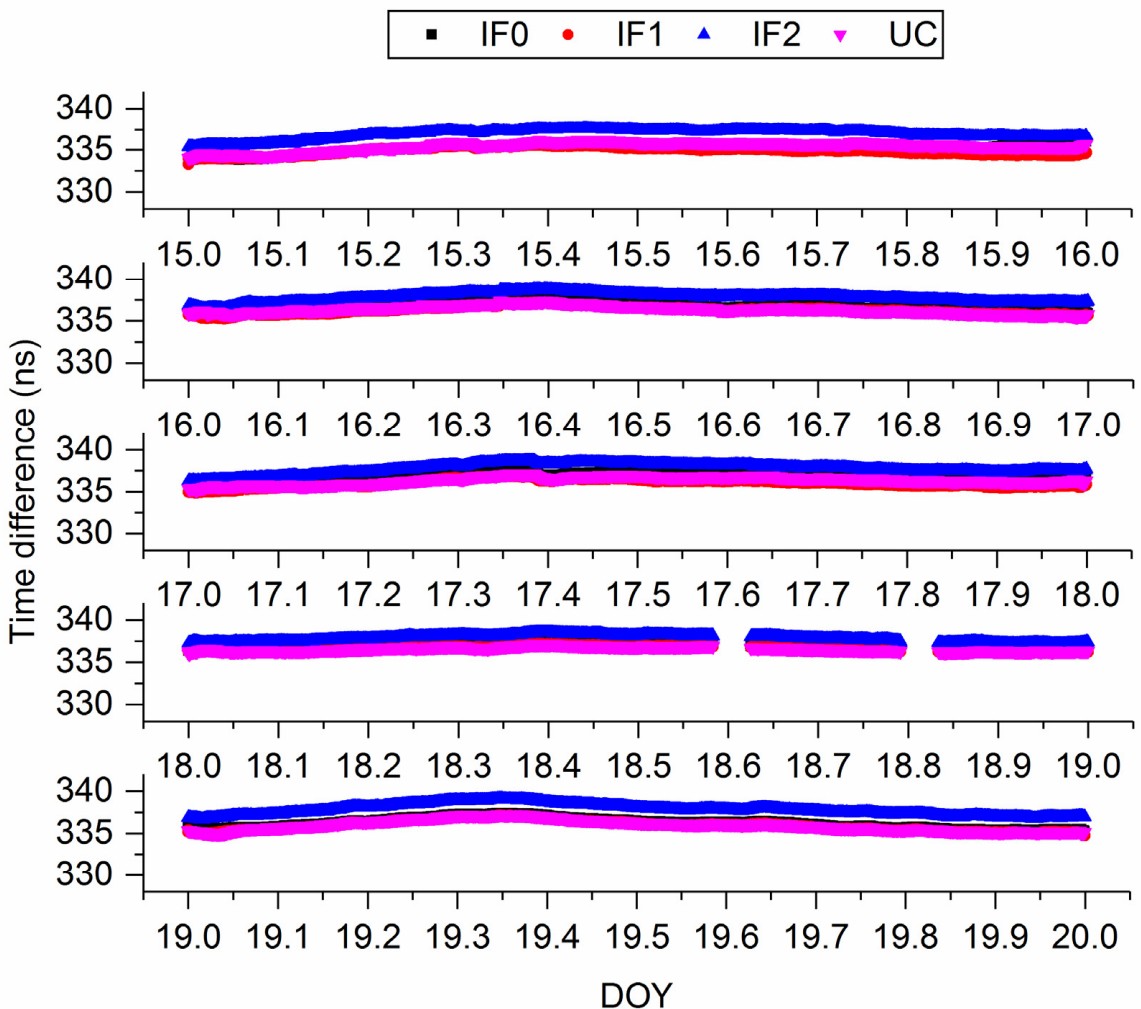

**Figure 3.** Comparison of time difference of XIA3-BRCH obtained from different models.

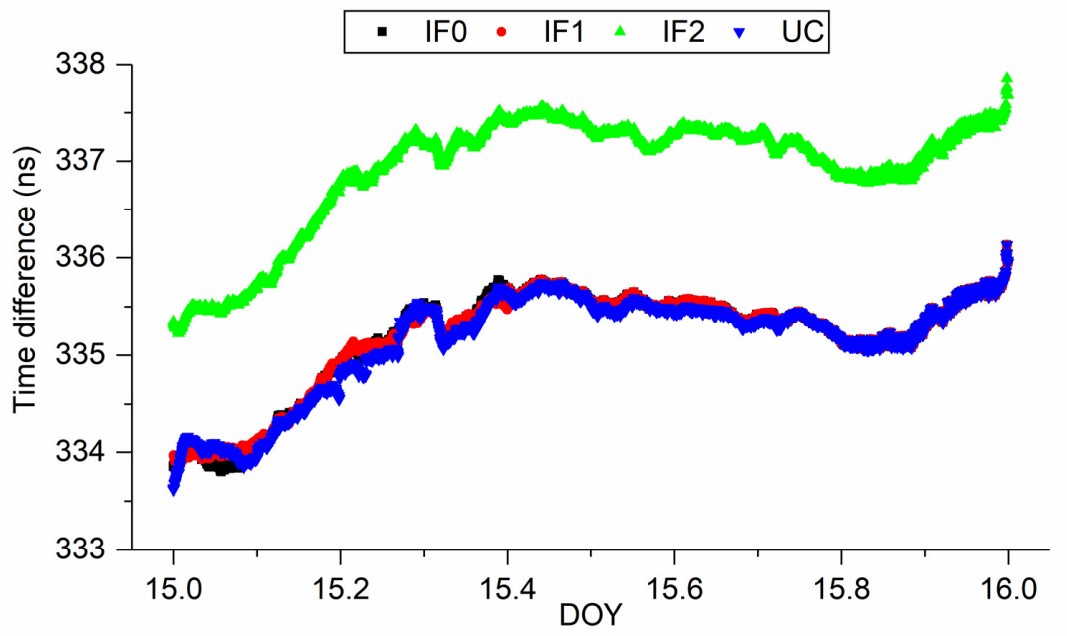

**Figure 4.** Comparison of time difference of XIA3-BRCH obtained from different models on DOY 15, 2019.

The frequency stability can be used to verify precise time transfer performance in another way. Here, the modified Allan deviation (MDEV) method is employed to assess the frequency stability. Figure 5 exhibits the modified Allan deviation (MDEV) values of the time differences, obtained from four kinds of BDS-3 precise time transfer models between BRCH and XIA3 stations at different time intervals. As shown in Figure 5, the MDEV of different precise time transfer is comparable to each other. At 30 s average time (frequency stability in short-term), the frequency stabilities are ($5.7869 \times 10^{-13}$, $4.5378 \times 10^{-13}$, $4.5589 \times 10^{-13}$, $4.8072 \times 10^{-13}$) for IF0, IF1, IF2, and UC model on DOY 15, 2019, respectively. Frequency stability are ($2.0471 \times 10^{-14}$, $2.589 \times 10^{-14}$, $1.9685 \times 10^{-14}$, $1.9147 \times 10^{-14}$) at 15,360 s average time (frequency stability in long-term) on DOY 15, 2019. We can conclude that the performance of quad-frequency PPP is equal to or better than that of dual-frequency PPP both for short- and long- term frequency stability. This finding can also be proved by Figure 6 (enlarged figure), which exhibits MDEV between different BDS−3 PPP models (DOY 15) on XIA3-BRCH time-link.

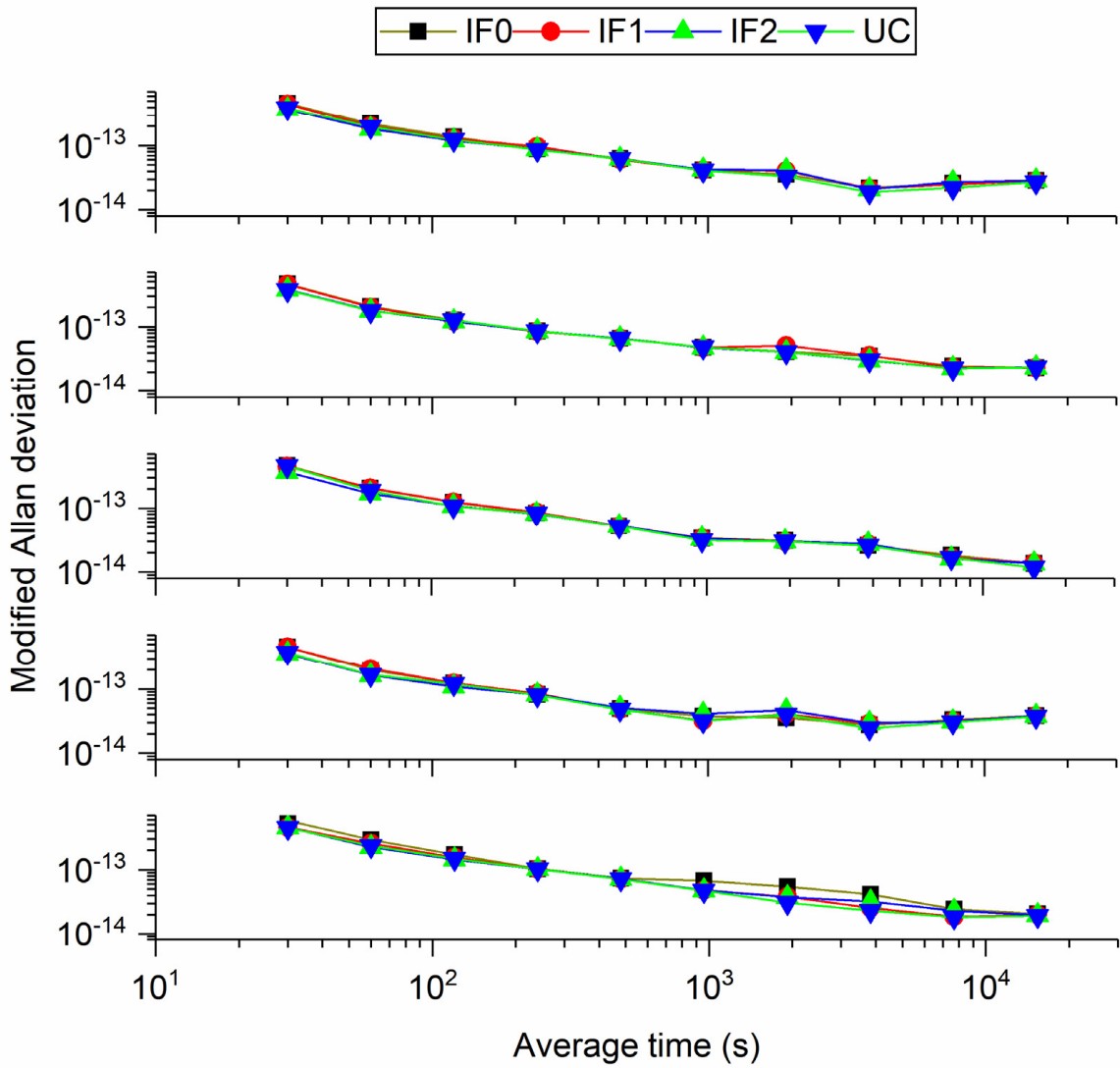

**Figure 5.** Comparison of MDEV between different BDS-3 PPP models from DOY 15 to 19 on XIA3-BRCH time-link.

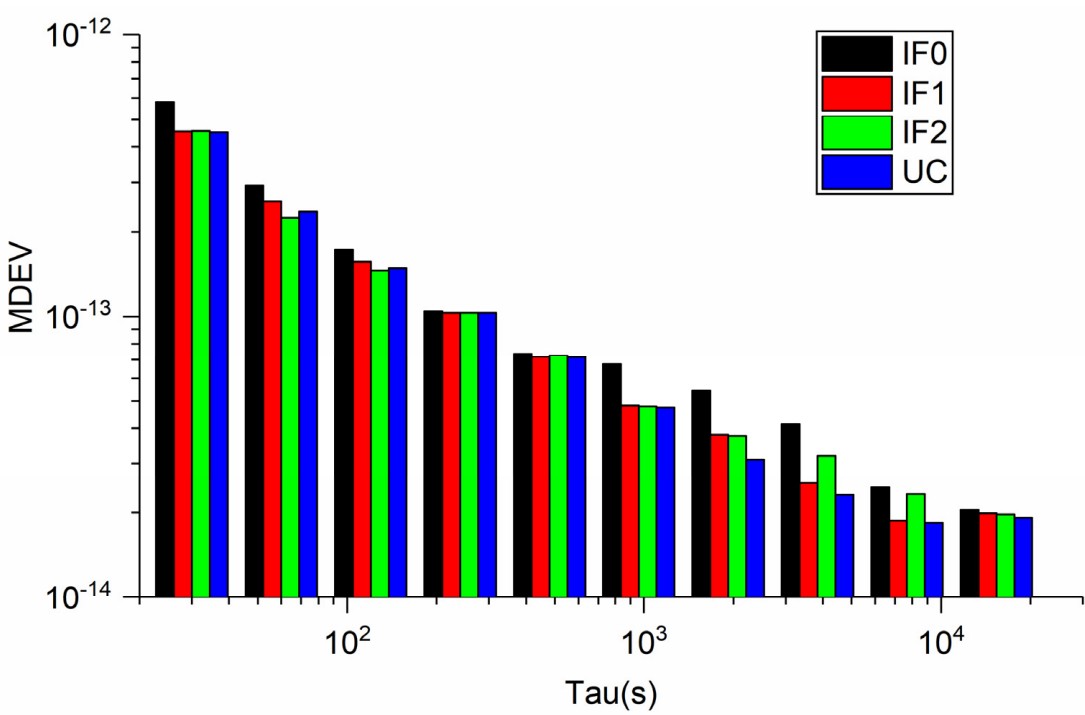

**Figure 6.** Comparison of MDEV between different BDS-3 PPP models (DOY 15) on XIA3-BRCH time-link.

### 3.3. Quad-Frequency Galileo PPP Time Transfer Solutions

Figures 7 and 8 present the time series of time difference obtained from different Galileo precise time transfer models on BRUX-PT11 and PTBB-PT11. In addition, time difference of BRUX-PT11 and PTBB-PT11 obtained from four models on DOY 277, 2019 is displayed in Figures 9 and 10, respectively, for clearly presentation. Note that the receivers at PT11 and PTBB are connected to the common clock at PTB. We can conclude two findings here. First, similar to the above conclusions, Galileo IF0, IF1, and UC agree well with each other. Second, IF2 time series is different from the other models, which is relative to the UCD. To further study the characteristic of Galileo precise time transfer model, we calculated mean and STD values of time series obtained from IF0, IF1, IF2, and UC models and listed in Tables 5 and 6. From DOY 227–282, the mean values are (−90.28, −90.46, −90.57, −90.77, −90.85, −90.79) ns, (−90.29, −90.42, −90.60, −90.77, −90.86, −90.79) ns, (−91.17, −91.34, −91.44, −91.65, −91.74, −91.68) ns, and (−90.27, −90.44, −90.57, −90.78, −90.87, −90.81) ns for IF0, IF1, IF2, and UC models, respectively, on BRUX-PT11 time-link. The STD values are (0.1, 0.03, 0.09, 0.05, 0.05, 0.06) ns, (0.09, 0.04, 0.11, 0.05, 0.06, 0.08) ns, (0.1, 0.03, 0.09, 0.05, 0.06, 0.07) ns, and (0.1, 0.03, 0.09, 0.05, 0.05, 0.06) ns, for IF0, IF1, IF2, and UC models, respectively, on BRUX-PT11 time-link. For a common clock experiment (PTBB-PT11), the mean values are (−146.6, −146.61, −146.6, −146.66, −146.66, −146.63) ns, (−146.67, −146.66, −146.65, −146.69, −146.72, −146.68) ns, (−144.6, −144.62, −144.59, −144.63, −144.7, −144.65) ns, for IF0, IF1, IF2, and UC models, respectively. The STD values are (0.03, 0.03, 0.03, 0.03, 0.03, 0.04) ns, (0.05, 0.04, 0.06, 0.04, 0.05, 0.06) ns, (0.04, 0.04, 0.05, 0.04, 0.05, 0.05) ns, (0.03, 0.03, 0.04, 0.03, 0.03, 0.04) ns for IF0, IF1, IF2, and UC models, respectively. Therefore, we obtain the second conclusion that quad-frequency Galileo precise time transfer models show the same performance. In addition, the quad-frequency Galileo PPP time transfer performance present a slightly better than that of BDS−3. There are several possible explanations for this result. First, the accuracy of BDS−3 orbit and clock products need to be further improved. Second, the TDOP values of Galileo satellite system is less than that of BDS-3. Third, the DCB values of BDS-3 we are currently used is just a preliminary file and has not yet been officially released. Besides, an unknown system bias exists in the IF2 model, so we do not recommend users to use this model for time transfer (See the mean values).

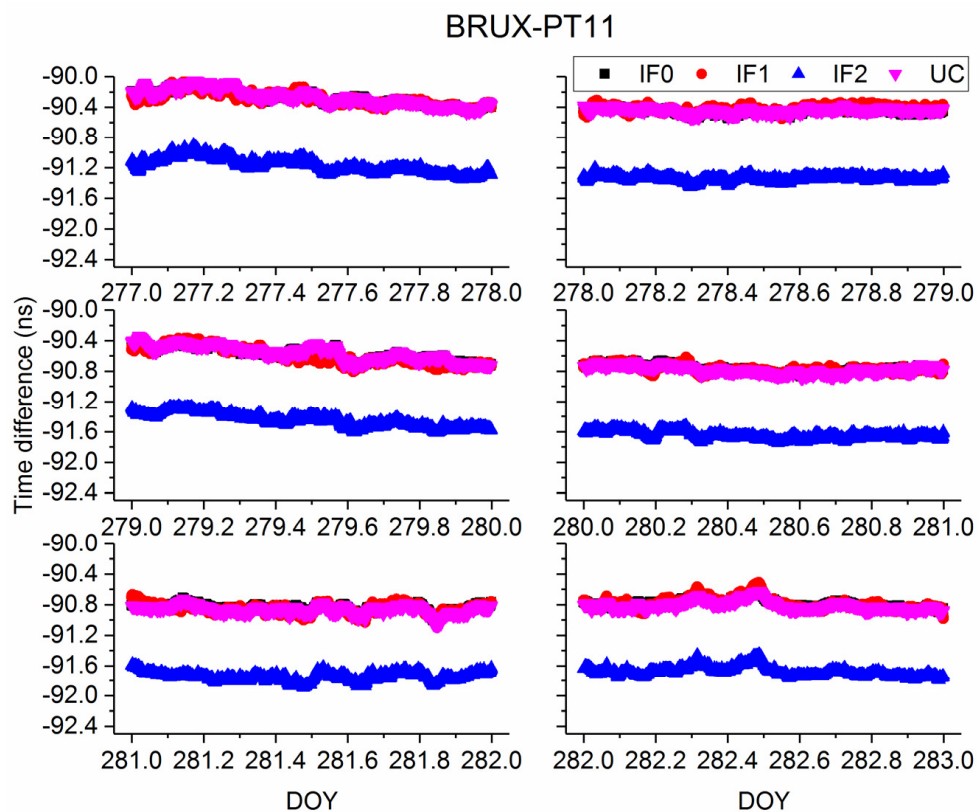

**Figure 7.** Comparison of time difference of BRUX-PT11 obtained from different models.

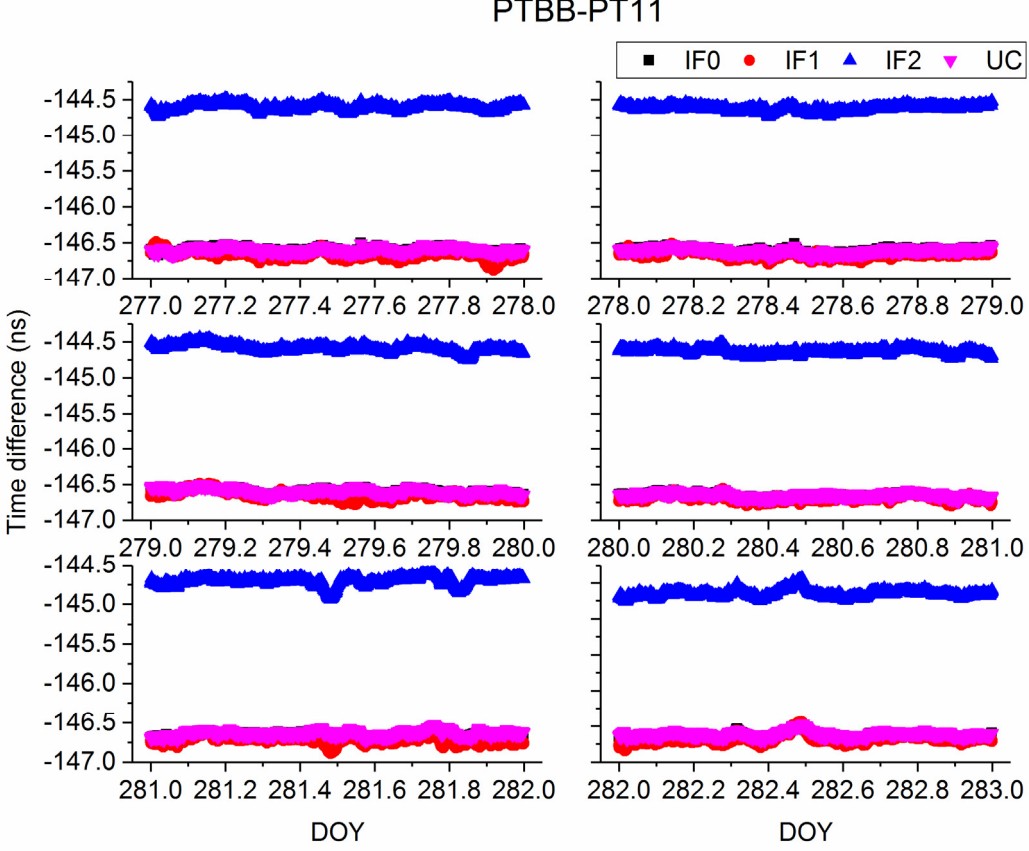

**Figure 8.** Comparison of time difference of PTBB-PT11 obtained from different models.

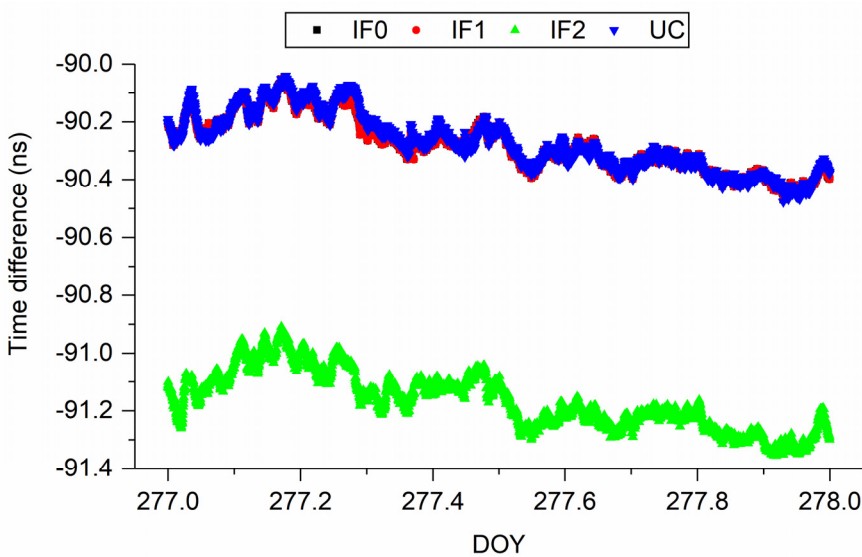

**Figure 9.** Comparison of time difference of BRUX-PT11 obtained from different models on DOY 277, 2019.

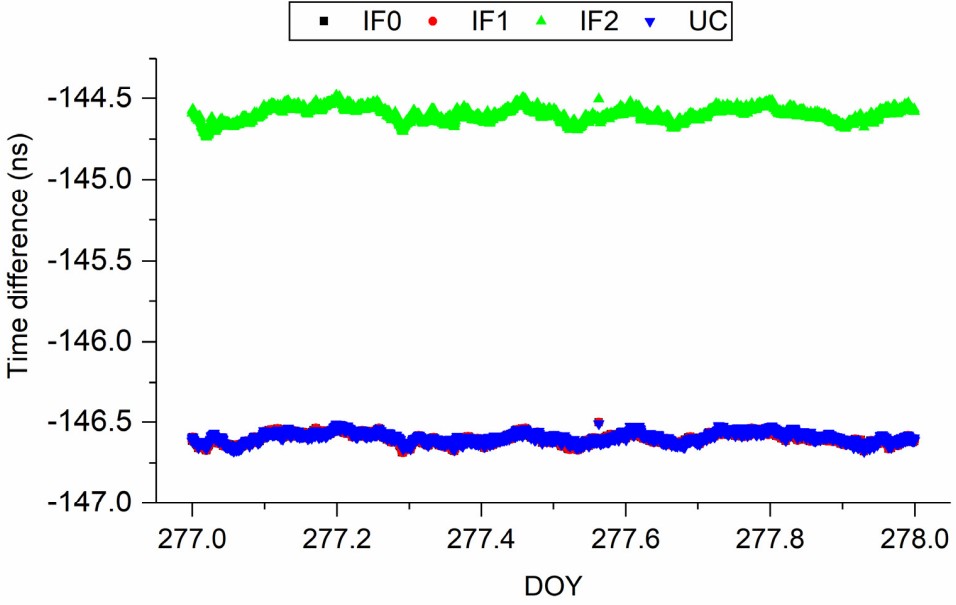

**Figure 10.** Comparison of time difference of PTBB-PT11 obtained from different models on DOY 277, 2019.

**Table 5.** Mean and STD values of time difference with different Galileo PPP models for BRUX-PT11 (ns).

| DOY | IF0 | | IF1 | | IF2 | | UC | |
|-----|------|-----|------|-----|------|-----|------|-----|
| | **Mean** | **STD** | **Mean** | **STD** | **Mean** | **STD** | **Mean** | **STD** |
| 277 | −90.28 | 0.10 | −90.29 | 0.09 | −91.17 | 0.10 | −90.27 | 0.10 |
| 278 | −90.46 | 0.03 | −90.42 | 0.04 | −91.34 | 0.03 | −90.44 | 0.03 |
| 279 | −90.57 | 0.09 | −90.60 | 0.11 | −91.44 | 0.09 | −90.57 | 0.09 |
| 280 | −90.77 | 0.05 | −90.77 | 0.05 | −91.65 | 0.05 | −90.78 | 0.05 |
| 281 | −90.85 | 0.05 | −90.86 | 0.06 | −91.74 | 0.06 | −90.87 | 0.05 |
| 282 | −90.79 | 0.06 | −90.79 | 0.08 | −91.68 | 0.07 | −90.81 | 0.06 |

**Table 6.** Mean and STD values of time difference with different Galileo PPP models for PTBB-PT11 (ns).

| DOY | IF0 | | IF1 | | IF2 | | UC | |
|---|---|---|---|---|---|---|---|---|
| | **Mean** | **STD** | **Mean** | **STD** | **Mean** | **STD** | **Mean** | **STD** |
| 277 | −146.60 | 0.03 | −146.67 | 0.05 | −144.60 | 0.04 | −146.60 | 0.03 |
| 278 | −146.61 | 0.03 | −146.66 | 0.04 | −144.62 | 0.04 | −146.61 | 0.03 |
| 279 | −146.60 | 0.03 | −146.65 | 0.06 | −144.59 | 0.05 | −146.59 | 0.04 |
| 280 | −146.66 | 0.03 | −146.69 | 0.04 | −144.63 | 0.04 | −146.65 | 0.03 |
| 281 | −146.66 | 0.03 | −146.72 | 0.05 | −144.70 | 0.05 | −146.64 | 0.03 |
| 282 | −146.63 | 0.04 | −146.68 | 0.06 | −144.65 | 0.05 | −146.61 | 0.04 |

Figures 11 and 12 display the MDEV of different models for BRUX-PT11 and PTBB-PT11 time-links. Obviously, the frequency stability is the same for four precise time transfer models. For the common clock experiment, actually, the MDEV of time series of PTBB-PT11 exhibits the noise and variations of the UCD and observations. We can see that the MDEV of six continuous days present the same variation. To show our results more clearly, Figures 13 and 14 present the MDEV of different models for BRUX-PT11 and PTBB-PT11 time-links on DOY 277, 2019. The frequency stability at 30 s and 15,360s outperforms $(3.9948 \times 10^{-13}, 3.9117 \times 10^{-13}, 3.9848 \times 10^{-13}, 3.92891 \times 10^{-13})$ and $(4.8298 \times 10^{-15}, 3.2336 \times 10^{-15}, 3.2903 \times 10^{-15}, 3.0274 \times 10^{-15})$ for BRUX-PT11 obtained from IF0, IF1, IF2, and UC model, respectively. In addition, the frequency stability at 30 s and 15,360 s are $(3.8044 \times 10^{-13}, 3.7898 \times 10^{-13}, 3.7668 \times 10^{-13}, 3.7603 \times 10^{-13})$ and $(1.4392 \times 10^{-15}, 1.4301 \times 10^{-15}, 1.4294 \times 10^{-15}, 1.3864 \times 10^{-15})$ for PTBB-PT11 obtained from IF0, IF1, IF2, and UC model, respectively. Obviously, we can see that quad-frequency Galileo has the same conclusions as quad-frequency BDS-3 PPP models.

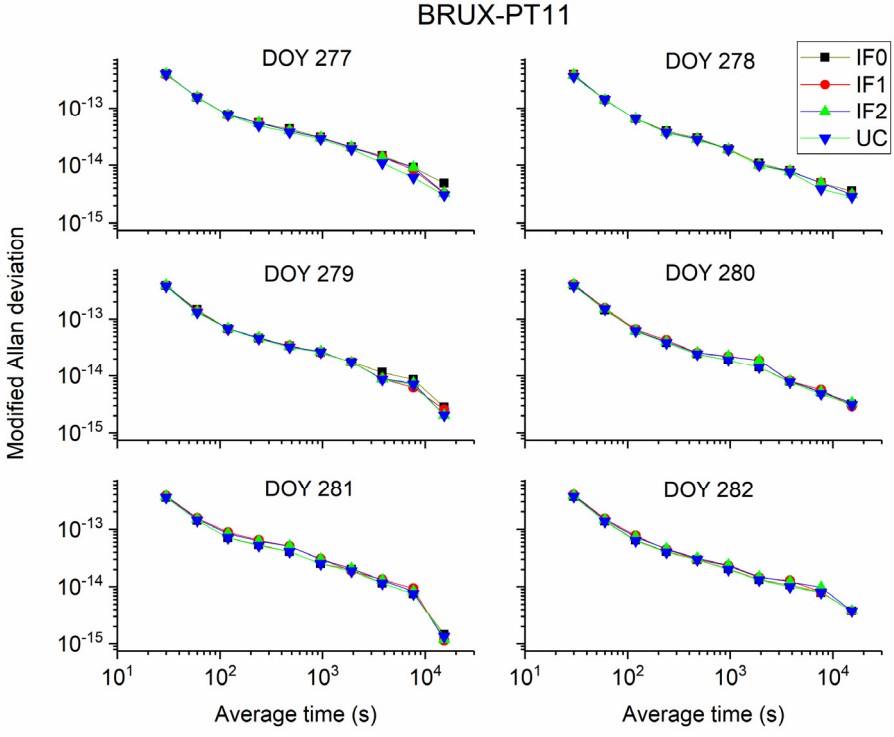

**Figure 11.** Comparison of MDEV between different Galileo PPP models for BRUX-PT11.

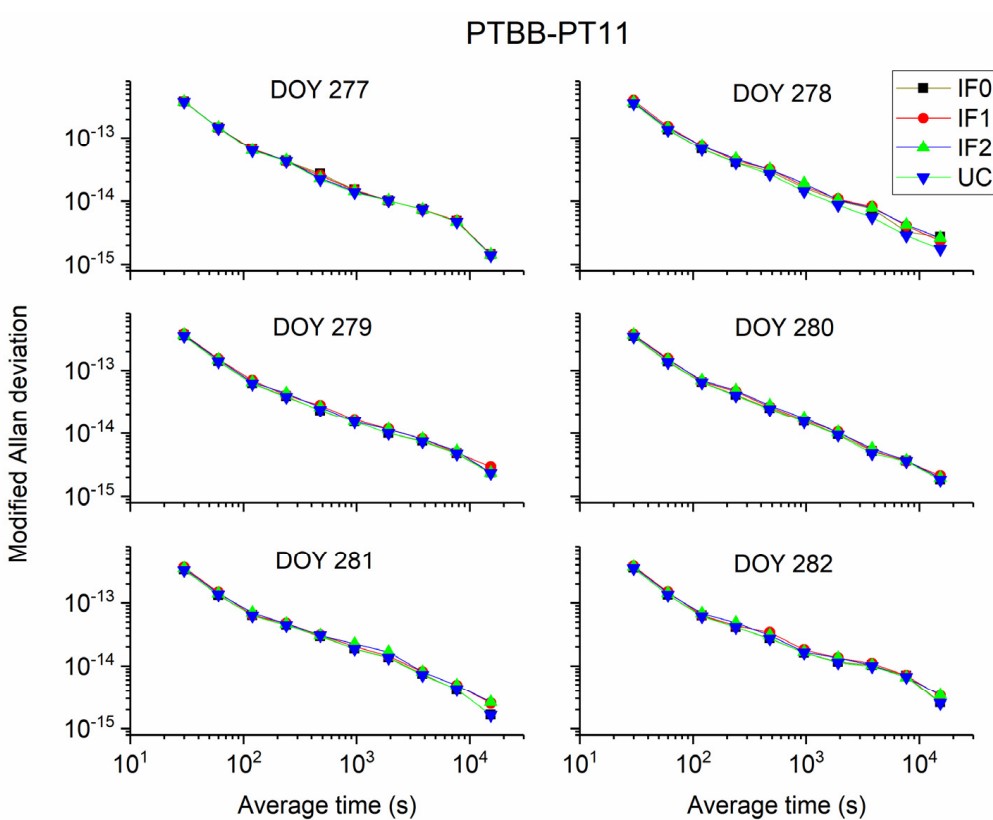

**Figure 12.** Comparison of MDEV between different Galileo PPP models for PTBB-PT11.

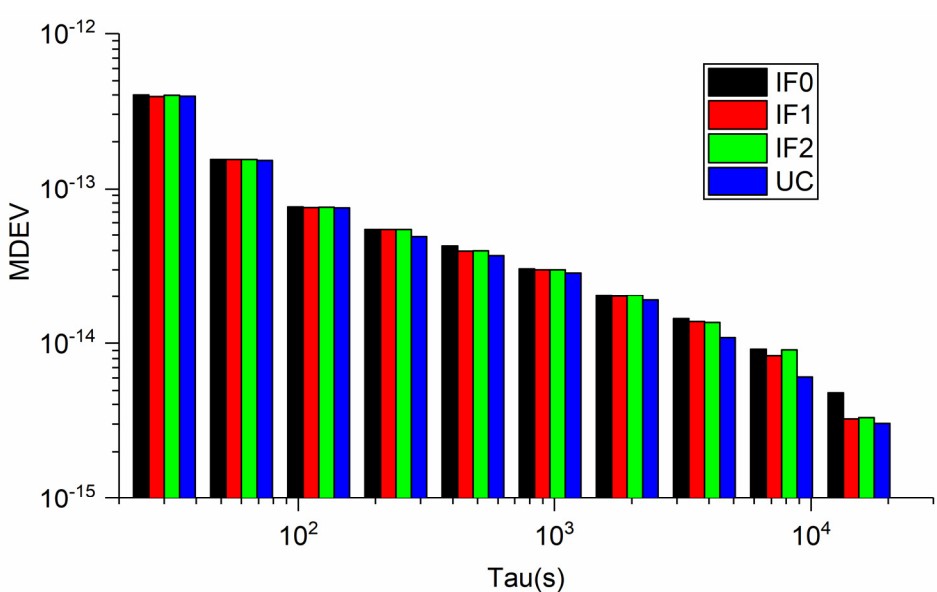

**Figure 13.** Comparison of MDEV between different Galileo PPP models for BRUX-PT11 on DOY 277, 2019.

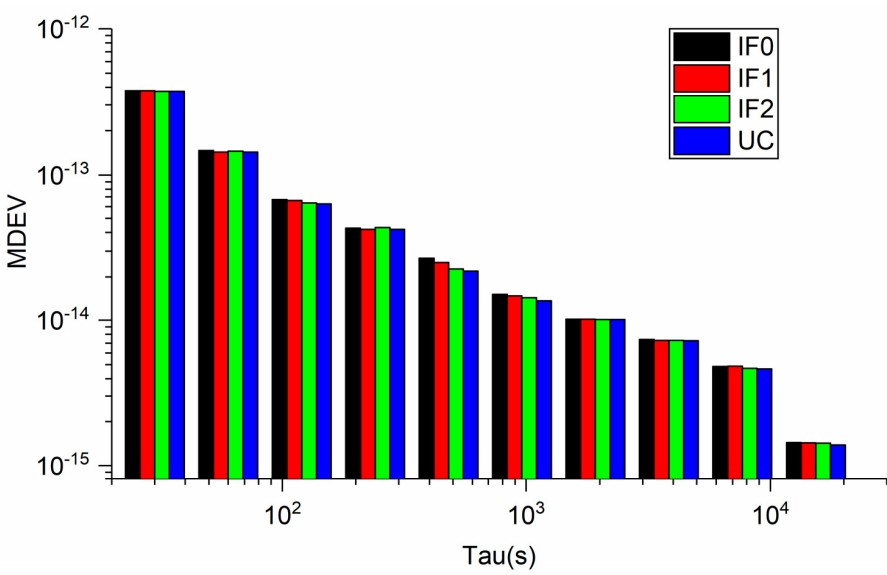

**Figure 14.** Comparison of MDEV between different Galileo PPP models for PTBB-PT11 on DOY 277, 2019.

### 3.4. Observation Residuals

Figure 15 illustrates the residuals of pseudorange observations at BRCH and XIA3 stations of BDS-3 IF0, IF1, IF2, and UC PPP models on DOY 16. Figure 16 demonstrates the residuals of pseudorange observations at BRUX and PT11 stations of Galileo IF0, IF1, IF2, and UC PPP models on DOY 278. Four findings can be concluded here. First, the pseudorange residuals are basically close to zero. It suggests that our DCB estimation and correction method is right. Second, the residuals of UC model are smaller than ionospheric-free models. It seems possible that these results are due to low noise amplification (see Tables 1 and 2). Third, for BDS-3 pseudorange residuals may be affected by the preliminary DCB file. Fourth, the pseudorange residuals of Galileo quad-frequency precise time transfer is obviously smaller than that of BDS-3. These differences can be explained by the fact that the orbit and clock products, DCB values and number of BDS-3 satellites need to be further improved.

### 3.5. Positioning, Tropospheric Delay, and IFB Estimates

For time users, the station coordinates are commonly estimated as a constant or constrained to improve receiver clock offset estimation. To fully study the quad-frequency precise time transfer method, the selected station coordinates were estimated as unknow parameters. Figures 17 and 18 exhibit the positioning error for different BDS-3 or Galileo PPP models on XIA3 and BRUX stations, respectively. Note that the precise receiver coordinates of PT11, BRCH, XIA3 are calculated by Bernese 5.2 software with IGS final products to assess the positioning accuracy. To further quantify positioning accuracy, the root mean square (RMS) of positioning error for BRCH and XIA3 stations with BDS-3 different PPP models and for BRUX, PT11, PTBB with Galileo different PPP models are listed in Tables 7 and 8. The mean RMS values of BDS-3 IF0, IF1, IF2, and UC models at BRCH station are (1.61, 0.46, 0.52, 0.77) cm, (0.78, 0.34, 0.63, 0.67) cm, and (2.54, 2.46, 2.40, 2.38) cm at east (E), north (N), up (U) components, respectively. At XIA3 station, the mean RMS values are (2.12, 2.22, 2.14, 1.93) cm, (0.61, 0.29, 0.34, 0.55) cm, and (3.45, 3.46, 3.35, 3.31) cm at E, N and U components. For Galileo precise time transfer models, the RMS values of IF0, IF1, IF2, and UC models at BRUX station are (0.39, 0.15, 0.17, 0.35) cm, (0.21, 0.21, 0.17, 0.11) cm, and (0.92, 0.96, 1.03, 0.88) cm at E, N and U components. For PTBB station, the mean RMS values are (0.26, 0.45, 0.17, 0.14) cm, (0.18, 0.23, 0.17, 0.17) cm, and (1.22, 0.87, 0.88, 0.78) cm at E, N and U components. Hence, both BDS-3 and Galileo

quad-frequency PPP time transfer models can reach few centimeters level about station coordinates parameters.

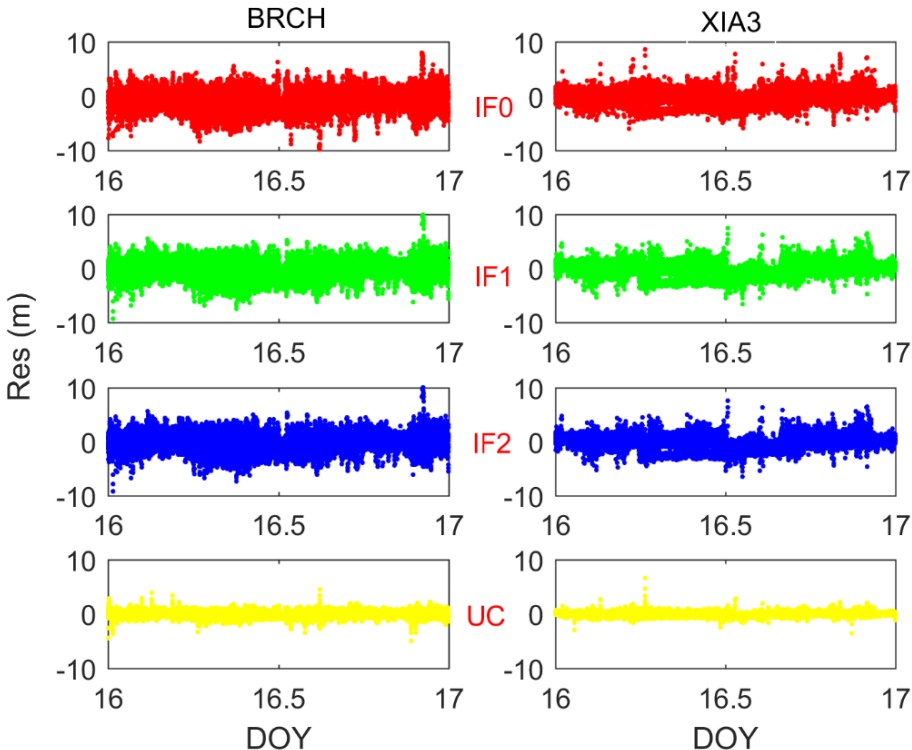

**Figure 15.** Analysis of the pseudorange observation residuals for all observed BDS-3 satellites for IF0, IF1, IF2, and UC model.

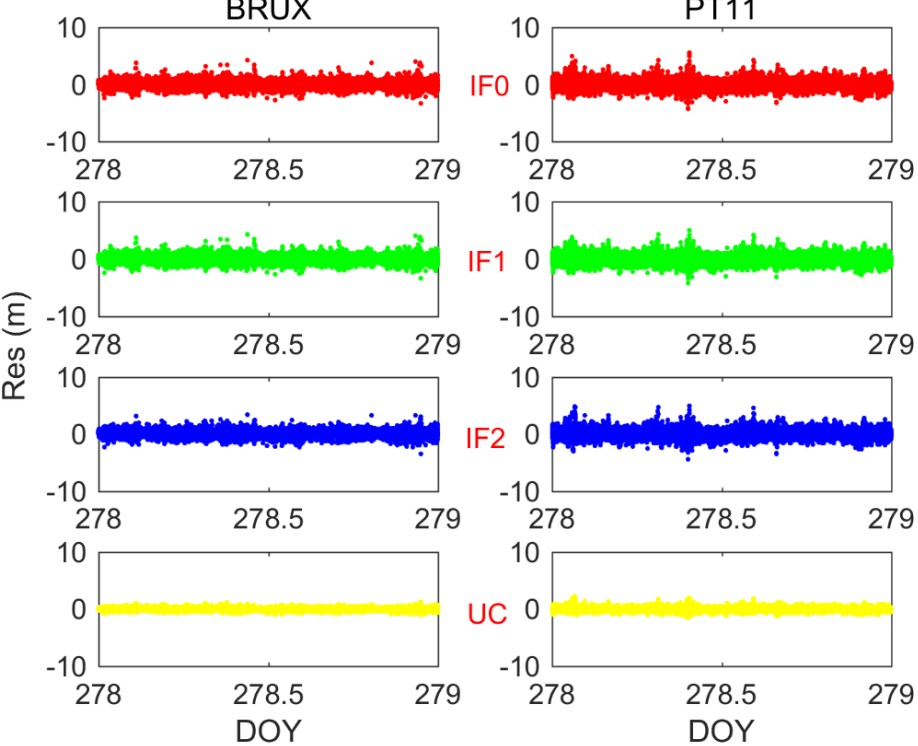

**Figure 16.** Analysis of the pseudorange observation residuals for all observed Galileo satellites for IF0, IF1, IF2, and UC models.

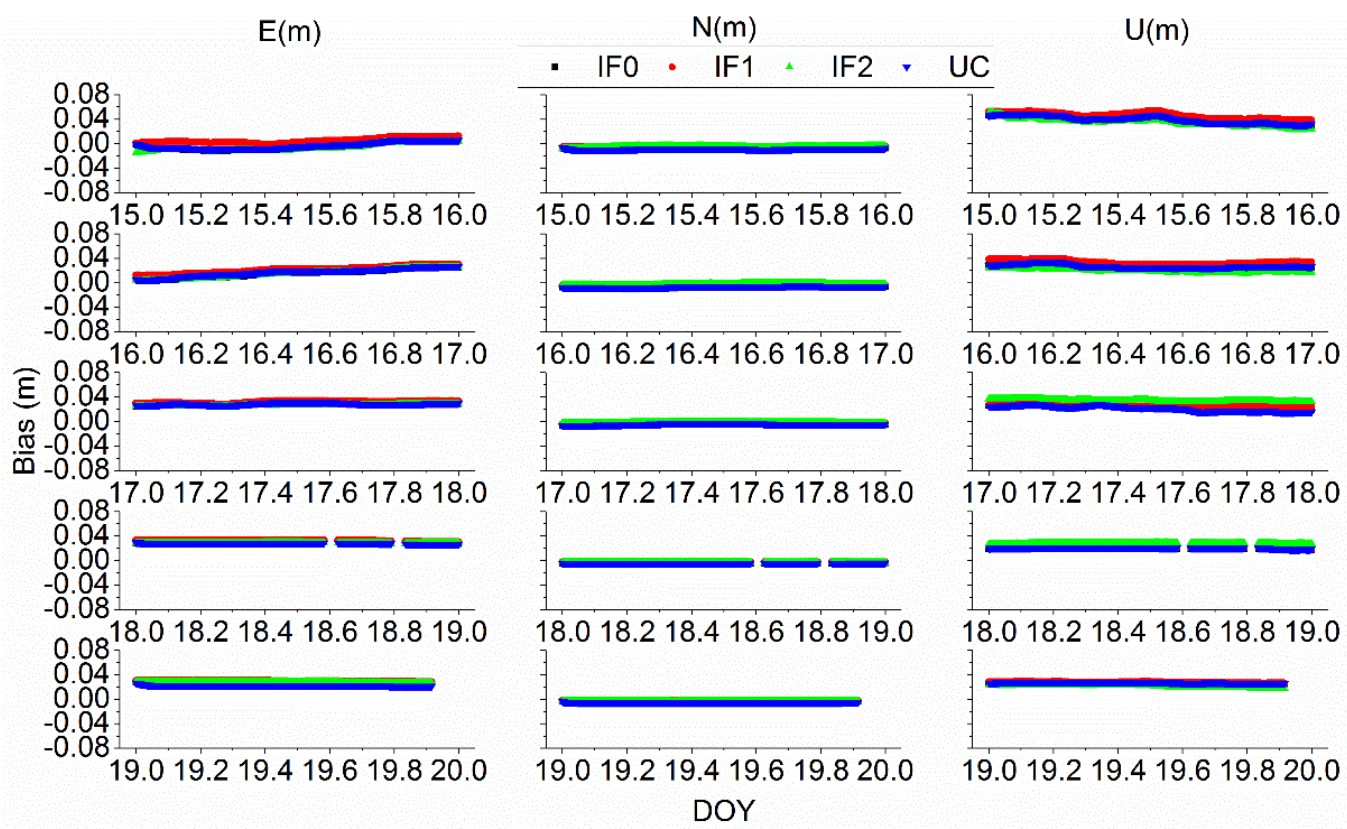

**Figure 17.** Position error between different BDS-3 PPP methods at XIA3.

**Table 7.** The RMS values of the position error for BRCH and XIA3 stations with different BDS-3 PPP models (cm).

|  | DOY | E (cm) | | | | N (cm) | | | | U (cm) | | | |
|---|---|---|---|---|---|---|---|---|---|---|---|---|---|
|  |  | IF0 | IF1 | IF2 | UC | IF0 | IF1 | IF2 | UC | IF0 | IF1 | IF2 | UC |
| BRCH | 15 | 2.03 | 0.71 | 0.65 | 1.03 | 0.51 | 0.29 | 0.59 | 0.59 | 2.76 | 2.25 | 2.72 | 2.13 |
|  | 16 | 1.84 | 0.56 | 0.41 | 0.56 | 0.77 | 0.31 | 0.69 | 0.48 | 2.87 | 2.46 | 2.67 | 2.79 |
|  | 17 | 1.67 | 0.41 | 0.70 | 0.82 | 0.92 | 0.43 | 0.77 | 0.75 | 2.10 | 2.67 | 2.14 | 2.05 |
|  | 18 | 1.64 | 0.30 | 0.64 | 0.90 | 0.84 | 0.31 | 0.61 | 0.67 | 2.54 | 2.52 | 2.34 | 2.50 |
|  | 19 | 0.85 | 0.30 | 0.22 | 0.55 | 0.85 | 0.35 | 0.51 | 0.84 | 2.42 | 2.38 | 2.15 | 2.41 |
| XIA3 | 15 | 0.75 | 0.67 | 0.90 | 0.59 | 0.93 | 0.50 | 0.52 | 0.84 | 3.91 | 3.68 | 3.48 | 3.70 |
|  | 16 | 1.59 | 2.25 | 1.61 | 1.91 | 0.71 | 0.38 | 0.27 | 0.62 | 2.53 | 3.39 | 2.91 | 2.49 |
|  | 17 | 2.79 | 2.75 | 2.62 | 2.03 | 0.45 | 0.19 | 0.22 | 0.41 | 3.35 | 3.53 | 3.00 | 3.20 |
|  | 18 | 2.86 | 2.81 | 2.84 | 2.86 | 0.52 | 0.19 | 0.33 | 0.43 | 3.85 | 3.10 | 3.79 | 3.57 |
|  | 19 | 2.63 | 2.60 | 2.75 | 2.25 | 0.45 | 0.18 | 0.35 | 0.43 | 3.61 | 3.60 | 3.57 | 3.57 |

For quad-frequency precise time transfer methods, the zenith troposphere delay parameter was treated as random walk noise. Figures 19 and 20 depict the troposphere delay estimated by BDS-3 or Galileo different PPP models, respectively. To further quantify the accuracy of troposphere delay parameter, the values obtained from Bernese 5.2 with IGS final products are further employed to assess troposphere delay. The RMS values are calculated and listed in Tables 9 and 10. The mean RMS values for BRCH and XIA3 are (1.10, 1.08, 1.09, 0.94) cm and (1.87, 1.97, 1.97, 1.91) cm for BDS-3 IF0, IF1, IF2, and UC PPP models. For Galileo IF0, IF1, IF2 and UC PPP models, the mean RMS values are (0.72, 0.86, 0.83, 0.69) cm, (0.67, 0.75, 0.72, 0.63) cm, and (0.68, 0.82, 0.82, 0.64) cm for BRUX, PT11, and PTBB stations. Thus, for BDS-3 and Galileo different models, there is no obvious difference for ZWD (see Tables 9 and 10).

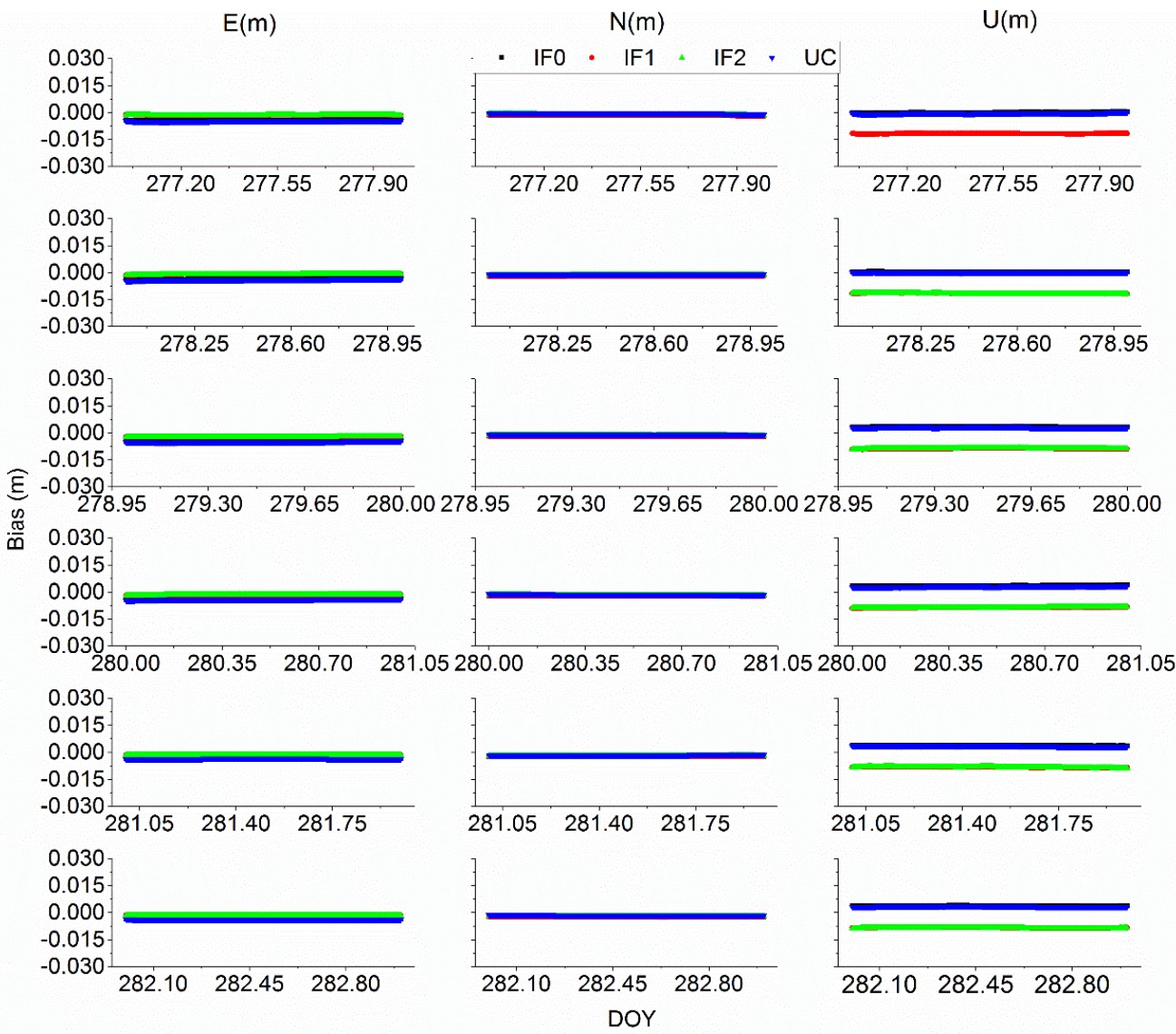

**Figure 18.** Position error between different Galileo PPP methods at BRUX.

**Table 8.** The RMS values of the position error for BRUX, PT11, and PTBB stations with different Galileo PPP models (ns).

| | DOY | E(cm) | | | | N(cm) | | | | U(cm) | | | |
|---|---|---|---|---|---|---|---|---|---|---|---|---|---|
| | | IF0 | IF1 | IF2 | UC | IF0 | IF1 | IF2 | UC | IF0 | IF1 | IF2 | UC |
| BRUX | 277 | 0.42 | 0.14 | 0.15 | 0.36 | 0.15 | 0.17 | 0.12 | 0.08 | 0.96 | 1.17 | 1.51 | 0.95 |
| | 278 | 0.35 | 0.09 | 0.11 | 0.35 | 0.19 | 0.20 | 0.15 | 0.08 | 0.99 | 1.16 | 1.18 | 0.91 |
| | 279 | 0.46 | 0.21 | 0.23 | 0.40 | 0.20 | 0.20 | 0.15 | 0.09 | 0.94 | 0.88 | 0.88 | 0.90 |
| | 280 | 0.39 | 0.16 | 0.17 | 0.35 | 0.23 | 0.22 | 0.18 | 0.13 | 0.86 | 0.86 | 0.87 | 0.81 |
| | 281 | 0.34 | 0.14 | 0.15 | 0.32 | 0.24 | 0.24 | 0.20 | 0.14 | 0.89 | 0.83 | 0.84 | 0.85 |
| | 282 | 0.35 | 0.16 | 0.18 | 0.31 | 0.24 | 0.25 | 0.20 | 0.14 | 0.88 | 0.85 | 0.87 | 0.85 |
| PT11 | 277 | 0.45 | 0.46 | 0.13 | 0.45 | 0.13 | 0.12 | 0.13 | 0.13 | 1.10 | 1.09 | 1.21 | 1.06 |
| | 278 | 0.41 | 0.42 | 0.11 | 0.40 | 0.11 | 0.08 | 0.11 | 0.11 | 0.90 | 0.88 | 1.00 | 0.77 |
| | 279 | 0.39 | 0.39 | 0.10 | 0.33 | 0.10 | 0.07 | 0.10 | 0.10 | 0.78 | 0.76 | 0.81 | 0.79 |
| | 280 | 0.34 | 0.34 | 0.08 | 0.37 | 0.08 | 0.06 | 0.08 | 0.08 | 0.80 | 0.80 | 0.82 | 0.75 |
| | 281 | 0.34 | 0.33 | 0.08 | 0.31 | 0.08 | 0.06 | 0.08 | 0.08 | 0.77 | 0.79 | 0.77 | 0.77 |
| | 282 | 0.35 | 0.34 | 0.12 | 0.31 | 0.12 | 0.06 | 0.12 | 0.12 | 0.80 | 0.83 | 0.83 | 0.73 |
| PTBB | 277 | 0.10 | 0.39 | 0.11 | 0.10 | 0.31 | 0.58 | 0.47 | 0.49 | 0.85 | 0.80 | 0.90 | 0.51 |
| | 278 | 0.36 | 0.80 | 0.34 | 0.26 | 0.13 | 0.21 | 0.12 | 0.11 | 1.12 | 0.82 | 0.92 | 0.73 |
| | 279 | 0.63 | 0.63 | 0.19 | 0.07 | 0.11 | 0.19 | 0.13 | 0.12 | 1.43 | 0.75 | 0.98 | 0.98 |
| | 280 | 0.20 | 0.52 | 0.20 | 0.18 | 0.15 | 0.18 | 0.13 | 0.09 | 1.28 | 0.99 | 0.89 | 0.82 |
| | 281 | 0.15 | 0.27 | 0.07 | 0.14 | 0.18 | 0.12 | 0.08 | 0.12 | 1.31 | 0.99 | 0.80 | 0.83 |
| | 282 | 0.10 | 0.08 | 0.11 | 0.07 | 0.19 | 0.09 | 0.08 | 0.09 | 1.33 | 0.89 | 0.78 | 0.82 |

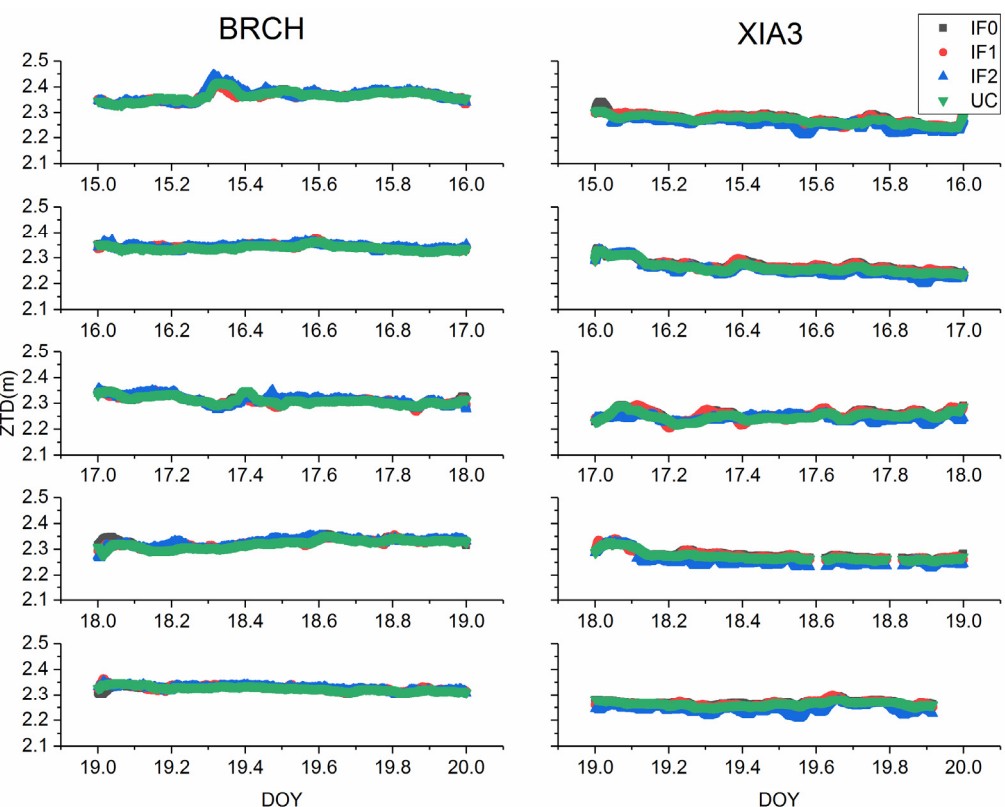

**Figure 19.** Zenith troposphere delay between different BDS-3 PPP methods.

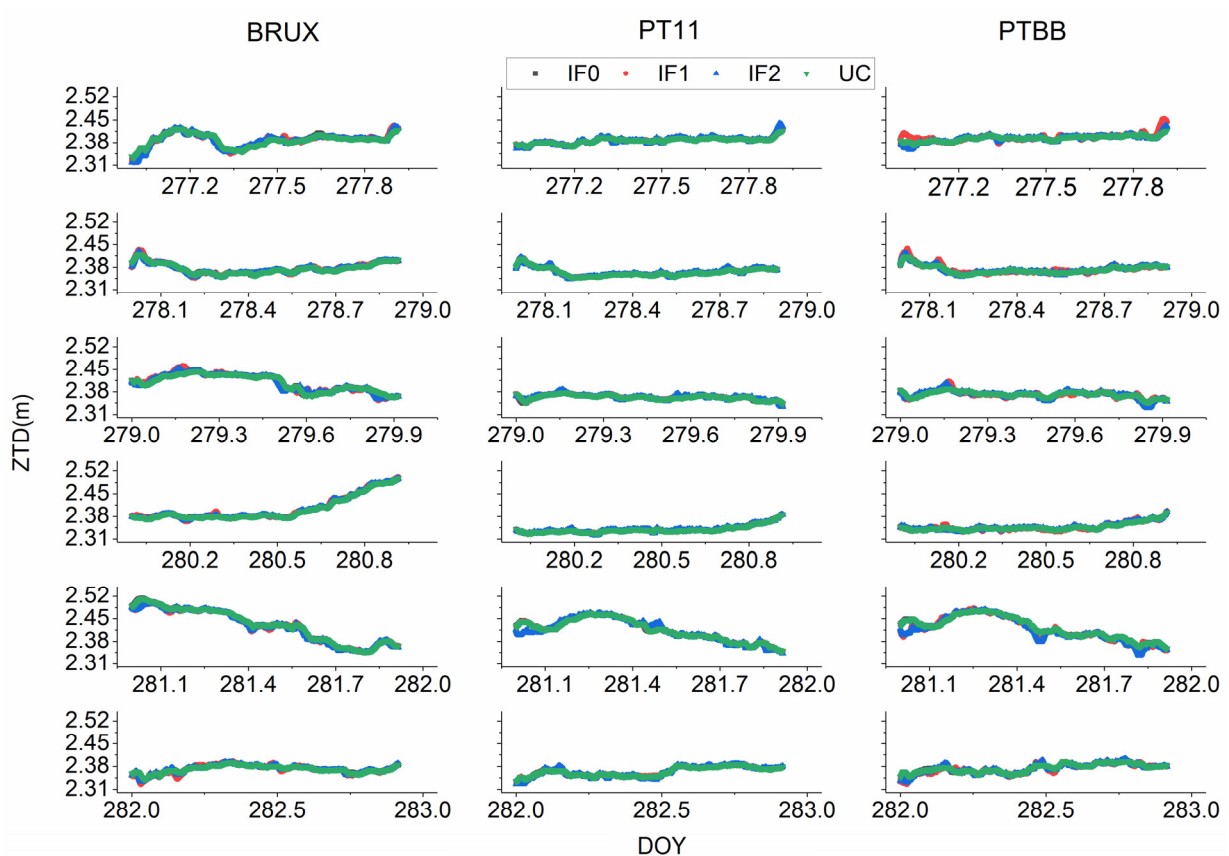

**Figure 20.** Zenith troposphere delay between different Galileo PPP methods.



**Table 9.** The RMS values of ZTD errors for BRCH and XIA3 stations with different BDS-3 PPP models (cm).

|  | DOY | IF0 | IF1 | IF2 | UC |
|---|---|---|---|---|---|
| BRCH | 15 | 1.3 | 1.32 | 1.32 | 1.31 |
|  | 16 | 0.80 | 0.80 | 0.83 | 0.84 |
|  | 17 | 1.20 | 1.40 | 1.40 | 1.21 |
|  | 18 | 1.20 | 1.02 | 1.02 | 0.64 |
|  | 19 | 1.00 | 0.88 | 0.88 | 0.74 |
| XIA3 | 15 | 1.76 | 1.82 | 1.82 | 1.81 |
|  | 16 | 2.06 | 2.11 | 2.11 | 2.01 |
|  | 17 | 2.17 | 2.20 | 2.19 | 2.01 |
|  | 18 | 1.66 | 1.93 | 1.94 | 1.95 |
|  | 19 | 1.72 | 1.78 | 1.78 | 1.78 |

**Table 10.** The RMS values of ZTD errors for BRUX, PT11, and PTBB stations with different Galileo PPP models (cm).

|  | DOY | IF0 | IF1 | IF2 | UC |
|---|---|---|---|---|---|
| BRUX | 277 | 0.98 | 1.00 | 1.04 | 0.88 |
|  | 278 | 0.61 | 0.76 | 0.70 | 0.57 |
|  | 279 | 0.60 | 0.92 | 0.76 | 0.64 |
|  | 280 | 0.52 | 0.63 | 0.63 | 0.52 |
|  | 281 | 0.83 | 1.04 | 0.96 | 0.81 |
|  | 282 | 0.77 | 0.80 | 0.86 | 0.71 |
| PT11 | 277 | 0.97 | 0.96 | 0.98 | 0.84 |
|  | 278 | 0.60 | 0.81 | 0.73 | 0.58 |
|  | 279 | 0.59 | 0.72 | 0.58 | 0.54 |
|  | 280 | 0.40 | 0.48 | 0.47 | 0.39 |
|  | 281 | 0.65 | 0.74 | 0.69 | 0.64 |
|  | 282 | 0.79 | 0.80 | 0.84 | 0.78 |
| PTBB | 277 | 0.86 | 0.90 | 0.91 | 0.72 |
|  | 278 | 0.53 | 0.63 | 0.62 | 0.50 |
|  | 279 | 0.75 | 0.88 | 0.90 | 0.70 |
|  | 280 | 0.53 | 0.68 | 0.57 | 0.51 |
|  | 281 | 0.8 | 0.91 | 1.05 | 0.83 |
|  | 282 | 0.60 | 0.90 | 0.87 | 0.56 |

For BDS-3 IF1, IF2 and UC model, there are one, one and two additional IFB parameters (see Equations (7)–(22)), which are related to the receiver after satellite DCB correction. Here, we will just show IFB parameters obtained from UC model at XIA3 and BRCH, which depicts at Figure 21. The mean STD values of IFB values are (0.018, 0.006) ns for BDS-3 UC models at BRCH and XIA3 stations, respectively. Hence, we can conclude that the IFB parameters are very stable during the whole day.

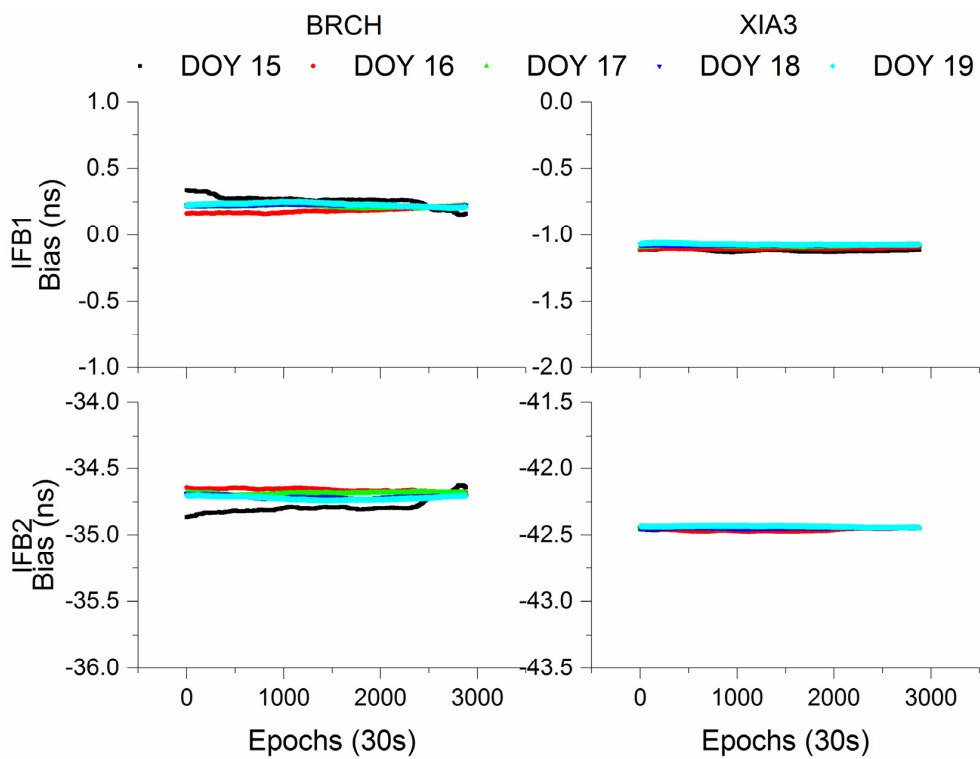

**Figure 21.** Analysis of two IFB parameters time series for BDS-3 UC models at BRCH and XIA3 stations.

For Galileo IF1, IF2, and UC models, two, one and two additional parameters also need to be estimated. As previous mentioned, Figure 22 exhibits the IFB parameters time series obtained from Galileo UC models. The mean STD values of IFB parameters are (0.014, 0.004) ns for UC models at BRUX and PT11 stations. Similar to the IFB parameters of BDS-3 PPP model, the IFB values vary stable, with the small STD values.

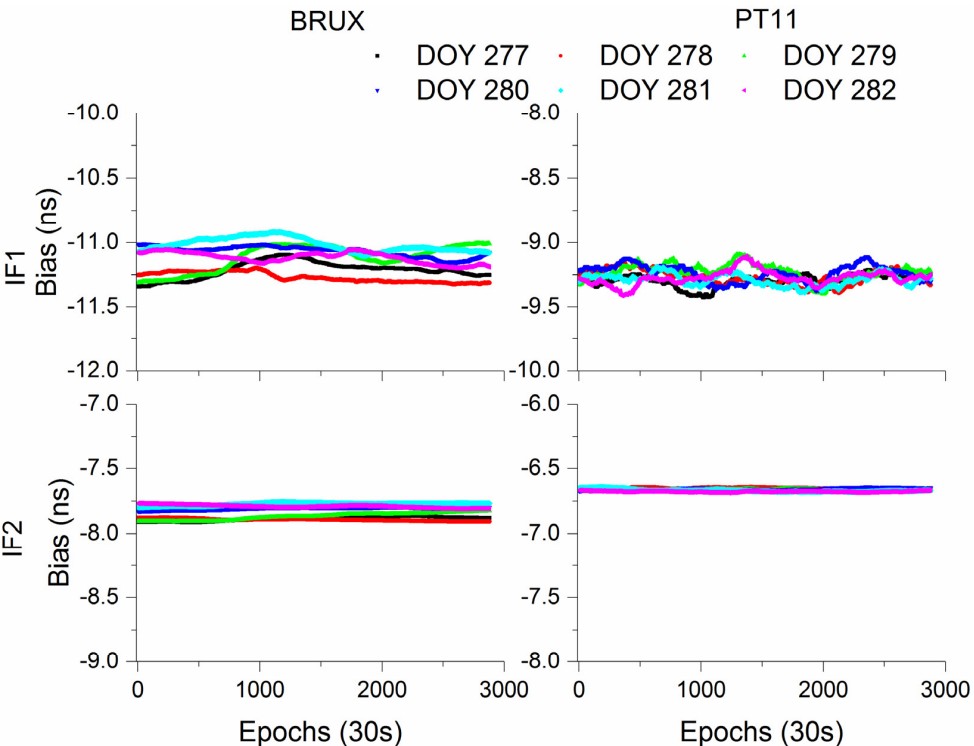

**Figure 22.** Analysis of two IFB parameters time series for Galileo UC model at BRUX and PT11 stations.

## 4. Conclusions

PPP time transfer with Galileo and BDS-3 quad-frequency observations was proposed in this work. Three quad-frequency PPP time and frequency transfer models, namely IF1, IF2, and UC, were studied and developed. The mathematical models of quad-frequency PPP were first described in detailed. Then, datasets from timing lab were selected and employed to evaluate reliability and feasibility of the proposed models. With the experimental results, three findings can be concluded.

First, the proposed three quad-frequency PPP models are still utilized to precise time transfer with BDS-3-only and Galileo-only quad-frequency observations. The reason for this is the fact that the frequency stability and time transfer accuracy are identical to or better than the corresponding IF0 PPP solutions.

Second, BDS-3-only or Galileo-only IF1, IF2, and UC models agree well with each other due to the same performance of positioning accuracy and tropospheric delay. However, we do not recommend time users to use IF2 models. This result may be explained by the fact that an unknown system bias exists in the IF2 model. Besides, IF2 model will be not used due to the absence of particular frequency. In addition, the receiver DCB values obtained from BDS-3 or Galileo quad-frequency PPP models vary stable.

Third, the Galileo quad-frequency PPP solutions perform slightly better than that of BDS-3. This result is explained by the fact that the orbit and clock products need to be further improved.

Currently, a few stations can receive all BDS-3 and Galileo quad-frequency signals. With the development of Galileo, BDS-3, system and GNSS tracking station, BDS-3 + Galileo combination PPP time and frequency with quad-frequency observations will be studied. In addition, PPP time and frequency transfer with ambiguity resolution of quad-frequency observations is of interest for future study. Unlike IF models, UC model provides not only station coordinates, receiver clock offset, troposphere delay, but also ionosphere information. Both BDS-3 or Galileo IF and UC models can be employed not only for positioning, troposphere delay, but also for precise time transfer.

**Author Contributions:** Y.G. and X.C. conceived and designed the experiments; Y.G. performed the experiments, analyzed the data, and wrote the paper; F.S., X.Y. and S.W. helped in the discussion and revision. All authors have read and agreed to the published version of the manuscript.

**Funding:** This work was supported by the National Natural Science Foundation of China (No. 41904018; 42077003), Natural Science Foundation of Jiangsu Province (No. BK20190714; BK20201374), High-level innovation and entrepreneurship talent plan of Jiangsu Province and K.C. Wong Education Foundation.

**Institutional Review Board Statement:** Not applicable.

**Informed Consent Statement:** Not applicable.

**Data Availability Statement:** The datasets analyzed in this study are managed by iGMAS and IGS.

**Acknowledgments:** The authors gratefully acknowledge iGMAS and IGS for providing broadcast ephemeris, data, and satellite information.

**Conflicts of Interest:** The authors declare no conflict of interest.

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
