# Peer review of "BDS-3/Galileo Time and Frequency Transfer with Quad-Frequency Precise Point Positioning"

_remotesensing, doi:10.3390/rs13142704_

Round 1

Reviewer 1 Report

The study proposed a method for quad-frequency precise point positioning (PPP) time and frequency transfer; however, the proposed method is not clear; the following comments may help to improve this study:

  • the stochastic models used and previous studies should be presented and discussed in the introduction. why author used stochastic should be discussed.
  • source of some equations should be cited, such as Eq.4, 7-10, and others.
  • the citation in the manuscript should be corrected in the whole manuscript.
  • the method section should be improved; the equations used in the manuscript is not clear and the study innovation and what authors did are also not presented and discussed; furthermore some results were presented without given a reasonable data or discussion. In addition, the authors also decided to used the elevation-dependent weight stochastic; however they didnot give the reason for their selection.
  • the results section includes a significant result; however, deep discussion is required for the presented figures; in addition, zoom-in can be used to show the results changes in the presented figures.

Author Response

Thank you very much for your encouragement and comments. We have revised the manuscript carefully and responded point by point to the comments as below. (C and R indicate comment and response, respectively). Our revisions are highlighted in our manuscript using the "Track Changes" function.

C: the stochastic models used and previous studies should be presented and discussed in the introduction. why author used stochastic should be discussed.

R: Thanks, According to your opinion, we have deleted this part.

C: source of some equations should be cited, such as Eq.4, 7-10, and others.

R: Accepted and modified.

C: the citation in the manuscript should be corrected in the whole manuscript.

R: Accepted and modified.

C: the method section should be improved; the equations used in the manuscript is not clear and the study innovation and what authors did are also not presented and discussed; furthermore some results were presented without given a reasonable data or discussion.

R: Thanks. The current studies mainly focused on dual- or triple-frequency PPP time transfer. The use of Galileo and BDS-3 quad-frequency observations for PPP time transfer present greater challenges. Therefore, Galileo and BDS-3 quad-frequency observations are anticipated to be useful in improving the performance of PPP time transfer. Our study innovation was presented in introduction. In our work, quad-frequency precise point positioning (PPP) time and frequency transfer methods using Galileo E1/E5a/E5b/E5 and BDS-3 B1I/B3I/B1C/B2a observations were proposed with corresponding mathematical models. In section 2, we give four quad-frequency PPP models in detail.

C: In addition, the authors also decided to used the elevation-dependent weight stochastic; however they didnot give the reason for their selection.

R: Thanks. The elevation-dependent weight stochastic is the common method, see“Tu, R.; Zhang, P.; Zhang, R.; Liu, J.; Lu, X. Modeling and performance analysis of precise time transfer based on BDS triple-frequency un-combined observations. Journal of Geodesy 2018, 93, 837-847, doi:10.1007/s00190-018-1206-3.”

C: the results section includes a significant result; however, deep discussion is required for the presented figures; in addition, zoom-in can be used to show the results changes in the presented figures.

R: Thank you for your suggestions. In the revision, we have added the figure with zoom-in and given deep discussion.

Author Response

(The authors gave the same response as above.)

Round 2

Reviewer 1 Report

the paper has been improved and can be accepted, thank you